# TOWARDS TRAINING WITHOUT DEPTH LIMITS:
## BATCH NORMALIZATION WITHOUT GRADIENT EXPLOSION

**Alexandru Meterez**[*]
D-INFK, ETH Zürich
ameterez@ethz.ch

**Amir Joudaki**[*]
D-INFK, ETH Zürich
ajoudaki@ethz.ch

**Francesco Orabona**
CEMSE, KAUST
francesco@orabona.com

**Alexander Immer**
D-INFK, ETH Zürich
aimmer@ethz.ch

**Gunnar Rätsch**
D-INFK, ETH Zürich
raetsch@inf.ethz.ch

**Hadi Daneshmand**
MIT LIDS/Boston University
hdanesh@mit.edu

## ABSTRACT

Normalization layers are one of the key building blocks for deep neural networks. Several theoretical studies have shown that batch normalization improves the signal propagation, by avoiding the representations from becoming collinear across the layers. However, results on mean-field theory of batch normalization also conclude that this benefit comes at the expense of exploding gradients in depth. Motivated by these two aspects of batch normalization, in this study we pose the following question: "Can a batch-normalized network keep the optimal signal propagation properties, but *avoid* exploding gradients?" We answer this question in the affirmative by giving a particular construction of an *Multi-Layer Perceptron (MLP) with linear activations* and batch-normalization that provably has *bounded gradients* at any depth. Based on Weingarten calculus, we develop a rigorous and non-asymptotic theory for this constructed MLP that gives a precise characterization of forward signal propagation, while proving that gradients remain bounded for linearly independent input samples, which holds in most practical settings. Inspired by our theory, we also design an activation shaping scheme that empirically achieves the same properties for certain non-linear activations.

## 1 INTRODUCTION

What if we could train even deeper neural networks? Increasing depth empowers neural networks, by turning them into powerful data processing machines. For example, increasing depth allows large language models (LLMs) to capture longer structural dependencies (Devlin et al., 2018; Liu et al., 2019; Brown et al., 2020; Goyal et al., 2021; Raffel et al., 2020). Also, sufficiently deep convolutional networks can outperform humans in image classification (Liu et al., 2022; Woo et al., 2023; Wu et al., 2021). Nevertheless, increasing depth imposes an inevitable computational challenge: deeper networks are harder to optimize. In fact, standard optimization methods exhibit a slower convergence when training deep neural networks. Hence, computation has become a barrier in deep learning, demanding extensive research and engineering.

A critical problem is that deeper networks suffer from the omnipresent issue of rank collapse at initialization: the outputs become collinear for different inputs as the network grows in depth (Saxe et al., 2013a). Rank collapse is not only present in MLPs and convolutional networks (Feng et al., 2022; Saxe et al., 2013a; Daneshmand et al., 2020), but also in transformer architectures (Dong et al., 2021; Noci et al., 2022). This issue significantly contributes to the training slowdown of deep neural networks. Hence, it has become the focus of theoretical and experimental studies (Saxe et al., 2013a; Feng et al., 2022; Daneshmand et al., 2021; Noci et al., 2022).

One of the most successful methods to avoid rank collapse is Batch Normalization (BN) (Ioffe & Szegedy, 2015), as proven in a number of theoretical studies (Yang et al., 2019; Daneshmand et al.,

---

[*]: Equal contribution
Code is available at: https://github.com/alexandrumeterez/bngrad

2020; 2021). Normalization imposes a particular bias across the layers of neural networks (Joudaki et al., 2023b). More precisely, the representations of a batch of inputs become more orthogonal after each normalization (Joudaki et al., 2023b). This orthogonalization effect precisely avoids the rank collapse of deep neural networks at initialization (Yang et al., 2019; Joudaki et al., 2023b; Daneshmand et al., 2021; Joudaki et al., 2023a).

While batch normalization effectively avoids rank collapse, it causes numerical issues. The existing literature proves that batch normalization layers cause exploding gradients in MLPs in an activation-independent manner (Yang et al., 2019). Gradient explosion limits increasing depth by causing numerical issues during backpropagation. For networks without batch normalization, there are effective approaches to avoid gradient explosion and vanishing, such as tuning the variance of the random weights based on the activation and the network width (He et al., 2015; Glorot & Bengio, 2010). However, such methods cannot avoid gradient explosion in the presence of batch normalization (Yang et al., 2019). Thus, the following important question remains unanswered:

*Is there any network with batch normalization without gradient explosion and rank collapse issues?*

**Contributions.** We answer the above question affirmatively by giving a specific MLP construction initialized with orthogonal random weight matrices, rather than Gaussian. To show that the MLP still has optimal signal propagation, we prove that the MLP output embeddings become isometric (equation 5), implying the output representations becomes more orthogonal with depth. For a batch of linearly independent inputs, we prove

$$\mathbb{E}\big[\text{isometry gap}\big] = \mathcal{O}\left(e^{-\text{depth}/C}\right), \tag{1}$$

where $C$ is a constant depending only on the network width and input and the expectation is taken over the random weight matrices. Thus, for sufficiently deep networks, the representations rapidly approach an orthogonal matrix. While Daneshmand et al. (2021) prove that the outputs converge to within an $\mathcal{O}(\text{width}^{-1/2})$-ball close to orthogonality, we prove that the output representations become perfectly orthogonal in the infinite depth limit. This perfect orthogonalization turns out to be key in proving our result about avoiding gradient explosion. In fact, for MLPs initialized with Gaussian weights and BN, Yang et al. (2019, Theorem 3.9) prove that the gradients explode at an exponential rate in depth. In a striking contrast, we prove that gradients of an MLP with BN and orthogonal weights remain bounded as

$$\mathbb{E}\big[\log\left(\text{gradient norm for each layer}\right)\big] = \mathcal{O}\left(\text{width}^5\right). \tag{2}$$

Thus, the gradient is bounded by a constant that only depends on the network width where the expectation is taken over the random weight matrices. It is worth noting that both isometry and log-norm gradient bounds are derived *non-asymptotically*. Thus, in contrast to the previously studied mean-field or infinite width regime, our theoretical results hold in practical settings where the width is finite.

The limitation of our theory is that it holds for a simplification in the BN module and linear activations. However, our results provide guidelines to avoid gradient explosion in MLPs with non-linear activations. We experimentally show that it is possible to avoid gradient explosion for certain non-linear activations with orthogonal random weights together with "activation shaping" (Martens et al., 2021). Finally, we experimentally demonstrate that avoiding gradient explosion stabilizes the training of deep MLPs with BN.

## 2 RELATED WORK

**The challenge of depth in learning.** Large depth poses challenges for the optimization of neural networks, which becomes slower by increasing the number of layers. This depth related slowdown is mainly attributed to: (i) gradient vanishing/explosion, and (ii) the rank collapse of hidden representations. (i) Gradient vanishing and explosion is a classic problem in neural networks (Hochreiter, 1998). For some neural architectures, this issue can be effectively solved. For example, He et al. (2015) propose a particular initialization scheme that avoids gradient vanishing/explosion for neural networks with rectifier non-linearities while Glorot & Bengio (2010) study the effect of initialization on sigmoidal activations. However, such initializations cannot avoid gradient explosion for

networks with batch normalization (Yang et al., 2019; Lubana et al., 2021). (ii) Saxe et al. (2013a) demonstrate that outputs become independent from inputs with growing depth, which is called the rank collapse issue (Daneshmand et al., 2020; Dong et al., 2021). Various techniques have been developed to avoid rank collapse such as batch normalization (Ioffe & Szegedy, 2015), residual connections (He et al., 2016), and self-normalizing activations (Klambauer et al., 2017). A related line of work has shown how signal propagation can be achieved without batch normalization in feed-forward networks (Burkholz & Dubatovka, 2019) and ResNets (Brock et al., 2021). Other works on CNNs (Blumenfeld et al., 2020) have shown that symmetry breaking is a vital element for achieving signal propagation with stable gradients in deep models. Here, we focus on batch normalization since our primary goal is to avoid the systemic issue of gradient explosion for batch normalization.

**Initialization with orthogonal matrices.** Saxe et al. (2013a) propose initializing the weights with random orthogonal matrices for linear networks without normalization layers. Orthogonal matrices avoid the rank collapse issue in linear networks, thereby enabling a depth-independent training convergence. Pennington et al. (2017) show that MLPs with sigmoidal activations achieve dynamical isometry when initialized with orthogonal weights. Similar benefits have been achieved by initializing CNNs with orthogonal or almost orthogonal kernels (Xiao et al., 2018; Mishkin & Matas, 2015), and by initializing RNN transition matrices with elements from the orthogonal and unitary ensembles (Arjovsky et al., 2016; Le et al., 2015; Henaff et al., 2016). Similarly, we use orthogonal random matrices to avoid gradient explosion. What sets our study apart from this literature is that our focus is on batch normalization and the issue of gradient explosion.

**Networks with linear activation functions.** Due to its analytical simplicity, the identity function has been widely used in theoretical studies for neural networks. Studies on identity activations date back to at least two decades. Fukumizu (1998) studies batch gradient descent in linear neural networks and its effect on overfitting and generalization. Baldi & Hornik (1995) provide an overview over various theoretical manuscripts studying linear neural networks. Despite linearity, as Saxe et al. (2013a;b) observe, the gradient dynamics in a linear MLP are highly nonlinear. In a line of work, Saxe et al. (2013b;a; 2019) study the training dynamics of deep neural networks with identity activations and introduce the notion of dynamical isometry. Baldi & Hornik (1989) and Yun et al. (2017) study the mean squared error optimization landscape in linear MLPs. More recently, the optimum convergence rate of gradient descent in deep linear neural networks has been studied by Arora et al. (2018) and Shamir (2019). Du & Hu (2019) prove that under certain conditions on the model width and input degeneracy, linear MLPs with Xavier initialized weights (Glorot & Bengio, 2010) converge linearly to the global optimum. Akin to these studies, we also analyze networks with linear activations. However, batch normalization is a non-linear function, hence the network we study in this paper is a highly non-linear function of its inputs.

## 3 MAIN RESULTS

We will develop our theory by constructing networks that do not suffer from gradient explosion (Sec. 3.3) and still orthogonalize (Sec. 3.1). The construction is similar to the network studied by Daneshmand et al. (2021): an MLP with batch normalization and linear activations. Formally, let $X^\ell \in \mathbb{R}^{d \times n}$ denote the representation of $n$ samples in $\mathbb{R}^d$ at layer $\ell$, then

$$X_{\ell+1} = \text{BN}(W_\ell X_\ell), \qquad \ell = 0, \ldots, L, \qquad (3)$$

where $W_\ell \in \mathbb{R}^{d \times d}$ are random weights, $n$ is the mini-batch size and $d$ is the feature dimension. Analogous to recent theoretical studies of batch normalization (Daneshmand et al., 2021; 2020), we define the BN operator $\text{BN} : \mathbb{R}^{d \times n} \to \mathbb{R}^{d \times n}$ as

$$\text{BN}(X) = \text{diag}(XX^\top)^{-\frac{1}{2}}X, \quad \text{BN}(X)_{ij} = \frac{X_{ij}}{\sqrt{\sum_{k=1}^d X_{ik}^2}} \,. \qquad (4)$$

Note that compared to the standard BN operator, mean reduction in equation 4 is omitted. Our motivation for this modification, similar to Daneshmand et al. (2021), is purely technical and to streamline our theory. We will experimentally show that using standard BN modules instead does not influence our results on gradient explosion and signal propagation (for more details see Figure H6).

A second minor difference is that in the denominator, we have omitted a $\frac{1}{n}$ factor. However, this only amounts to a constant scaling of the representations and does not affect our results.

Compared to Daneshmand et al. (2021), we need two main modifications to avoid gradient explosion: (i) $n = d$, and (ii) $W_\ell$ are random *orthogonal* matrices. More precisely, we assume the distribution of $W_\ell$ is the Haar measure over the orthogonal group denoted by $\mathbb{O}_d$ (Collins & Śniady, 2006). Such an initialization scheme is widely used in deep neural networks without batch normalization (Saxe et al., 2013a; Xiao et al., 2018; Pennington et al., 2017). For MLP networks with BN, we prove such initialization avoids the issue of gradient explosion, while simultaneously orthogonalizing the inputs.

### 3.1 TRACKING SIGNAL PROPAGATION VIA ORTHOGONALITY

As discussed, batch normalization has an important orthogonalization bias that influences training. Without normalization layers, representations in many architectures face the issue of rank-collapse, which happens when network outputs become collinear for arbitrary inputs, hence their directions become insensitive to the changes in the input. In contrast, the outputs in networks with batch normalization become increasingly orthogonal through the layers, thereby enhancing the signal propagation in depth (Daneshmand et al., 2021). Thus, it is important to check whether the constructed network maintains the important property of orthogonalization.

**Isometry gap.** Our analysis relies on the notion of *isometry gap*, $\phi : \mathbb{R}^{d \times n} \to \mathbb{R}$, introduced by Joudaki et al. (2023b). Isometry gap is defined as

$$\phi(X) = -\log\left(\frac{\det(X^\top X)^{\frac{1}{d}}}{\frac{1}{d}\text{Tr}(X^\top X)}\right). \tag{5}$$

One can readily check that $\phi(X) \geq 0$ and it is zero when $X$ is an orthogonal matrix, i.e., $XX^\top = I_d$. The *isometry* denoted by $\mathcal{I} : \mathbb{R}^{d \times n} \to \mathbb{R}$ is defined as $\mathcal{I}(X) = \exp(-\phi(X))$.

**Geometric interpretation of isometry.** While the formula for isometry gap may seem enigmatic at first, it has a simple geometric interpretation that makes it intuitively understandable. The determinant $\det(X^\top X) = \det(X)^2$ is the squared volume of the parallelepiped spanned by the columns of $X$, while $\text{Tr}(X^\top X)$ is the sum squared-norms of the columns of $X$. Thus, the ratio between the two provides a scale-invariant notion of volume and isometry. On the one hand, if there is any collinearity between the columns, the volume will vanish and the isometry gap will be infinity, $\phi(X) = \infty$. On the other hand, $\phi(X) = 0$ implies $X^\top X$ is a scaled identity matrix. We will prove $\phi$ serves as a Lyapunov function for the chain of hidden representations $\{X_\ell\}_{\ell=0}^\infty$.

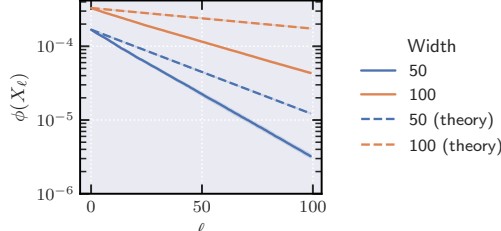

Figure 1: Isometry gap (y-axis, log-scale) in depth for an MLP with orthogonal weights, over randomly generated data. As predicted by Theorem 1, isometry gap of representations vanishes at an exponential rate. The solid traces are averaged over 10 independent runs, and the dashed traces show the theoretical prediction from Theorem 1.

**Theory for orthogonalization.** The following theorem establishes the link between orthogonality of representations and depth.

**Theorem 1.** *There is an absolute constant $C$ such that for any layer $\ell \leq L$ we have*

$$\mathbb{E}\phi(X_{\ell+1}) \leq \phi(X_0)e^{-\ell/k}, \qquad \text{where} \qquad k := Cd^2(1 + d\phi(X_0)). \tag{6}$$

Theorem 1 states that if the samples in the input batch are not linearly dependent, representations approach orthogonality at an exponential rate in depth. The orthogonalization in depth ensures the avoidance of the rank collapse of representations, which is a known barrier to training deep neural networks (Daneshmand et al., 2020; Saxe et al., 2013a; Bjorck et al., 2018).

Figure 1 compares the established theoretical decay rate of $\phi$ with the practical rate. Interestingly, the plot confirms that the rate depends on width in practice, akin to the theoretical rate in Theorem 1. It is worth mentioning that the condition on the input samples to not be linearly dependent is necessary to establish this result. One can readily check that starting from a rank-deficient input, neither matrix products, nor batch-normalization operations can increase the rank of the representations. Since this assumption is quantitative, we can numerically verify it by randomly drawing many input mini-batches and check if they are linearly dependent. For CIFAR10, CIFAR100, MNIST and FashionMNIST, we empirically tested that most batches across various batch sizes are full-rank (see Section E for details on the average rank of a batch in these datasets).

Theorem 1 distinguishes itself from the existing orthogonalization results in the literature (Yang et al., 2019; Joudaki et al., 2023b) as it is non-asymptotic and holds for networks with finite width. Since practical networks have finite width and depth, non-asymptotic results are crucial for their applicability to real-world settings. While Daneshmand et al. (2021) provide a non-asymptotic bound for orthogonalization, the main result relies on an assumption that is hard to verify.

*Proof idea of Theorem 1.* We leverage a recent result established by Joudaki et al. (2023b), proving that the isometry gap does not decrease with BN layers. For all non-degenerate matrices $X \in \mathbb{R}^{d \times d}$, the following holds

$$\mathcal{I}(\text{BN}(X)) \geq \left(1 + \frac{\text{variance}\{\|X_{j\cdot}\|\}_{j=1}^d}{(\text{mean}\{\|X_{j\cdot}\|\}_{j=1}^d)^2}\right) \mathcal{I}(X) .$$

Using the above result, we can prove that matrix multiplication with orthogonal weights also does not decrease isometry as stated in the next lemma.

**Lemma 2** (Isometry after rotation). *Let $X \in \mathbb{R}^{d \times d}$ and $W \in \mathbb{R}^{d \times d}$ be an orthogonal matrix and $X' = WX$; then,*

$$\mathcal{I}(\text{BN}(X')) \geq \left(1 + \frac{\text{variance}\{\|X'_{j\cdot}\|\}_{j=1}^d}{(\text{mean}\{\|X'_{j\cdot}\|\}_{j=1}^d)^2}\right) \mathcal{I}(X) . \tag{7}$$

It is straightforward to check that there exists at least an orthogonal matrix $W$ for which $\mathcal{I}(\text{BN}(WX)) = 1$ (see Corollary B.3). Thus, $\mathcal{I}(\cdot)$ strictly increases for some weight matrices, as long as $X$ is not orthogonal. When the distribution of $W$ is the Haar measure over the orthogonal group, we can leverage recent developments in Weingarten calculus (Weingarten, 1978; Banica et al., 2011; Collins & Śniady, 2006; Collins et al., 2022) to calculate a rate for the isometry increase in expectation:

**Theorem 3.** *Suppose $W \sim \mathbb{O}_d$ is a matrix drawn from $\mathbb{O}_d$ such that the distribution of $W$ and $UW$ are the same for all orthogonal matrices $U$. Let $\{\lambda_i\}_{i=1}^d$ be the eigenvalues of $XX^\top$. Then,*

$$\mathbb{E}_W\left[\mathcal{I}(\text{BN}(WX))\right] \geq \left(1 - \frac{\sum_{k=1}^d(\lambda_k - 1)^2}{2d^2(d+2)}\right)^{-1} \mathcal{I}(X) \tag{8}$$

*holds for all $X = \text{BN}(\cdot)$, with equality for orthogonal matrices.*

The structure in $X$ induced by BN ensures its eigenvalues lie in the interval $(0, 1]$, in that the multiplicative factor in the above inequality is always greater than one. In other words, $\mathcal{I}(\cdot)$ increases by a constant factor in expectation that depends on how close $X$ is to an orthogonal matrix.

The connection between Theorem 3 and the main isometry gap bound stated in Theorem 1 is established in the following Corollary (recall $\phi = -\log \mathcal{I}$).

**Corollary 4** (Isometry gap bound). *Suppose the same setup as in Theorem 3, where $X' = WX$. Then, we have:*

$$\mathbb{E}_W[\phi(X')|X] \leq \phi(X) + \log\left(1 - \frac{\sum_k(\lambda_k - 1)^2}{2d^2(d+2)}\right) . \tag{9}$$

Notice that the term $\frac{\sum_{k=1}^d(\lambda_k-1)^2}{2d^2(d+2)} = \mathcal{O}(\frac{1}{d})$, yielding $\log\left[1 - \frac{\sum_{k=1}^d(\lambda_k-1)^2}{2d^2(d+2)}\right] \leq 0$.

The rest of proof is based on an induction over the layers, presented in Appendices B and C.

## 3.2 ORTHOGONALIZATION AND GRADIENT EXPLOSION

There is a subtle connection between orthogonalization and gradient explosion. Suppose the input batch is rank-deficient, i.e., degenerate. As elaborated above, since all operations in our MLP can be formulated as matrix products, they cannot recover the rank of the representations, which thus remain degenerate. By perturbing the input such that it becomes full-rank, the output matrix becomes orthogonal, hence non-degenerate at an exponential rate in depth as proven in Theorem 1.

Thus, a slight change in inputs leads to a significant change in outputs from degeneracy to orthogonality. Considering that the gradient measures changes in the loss for infinitesimal inputs changes, the large changes in outputs potentially lead to gradient explosion. While this is only an intuitive argument, we observe that in practice the gradient does explode for degenerate inputs, as shown in Figure 2.

Nonetheless, in Figure 2 we observe that for non-degenerate inputs the gradient norm does not explode. In fact, we observe that inputs are often non-degenerate in practice (see Table E1 for details). Thus, an important question is whether the gradient norm remains bounded for non-degenerate input batches. Remarkably, we can not empirically verify that for *all degenerate inputs* the gradient norm remains bounded. Therefore, a theoretical guarantee is necessary to ensure avoiding gradient explosion.

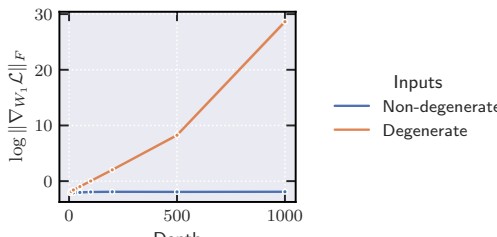

Figure 2: Logarithmic plot for the gradient norm of the first layer for networks with different number of layers evaluated on degenerate (orange) and non-degenerate (blue) inputs. The degenerate inputs contain repeated samples from CIFAR10 in the batch, measured at initialization for MLPs of various depths. While gradients explode for degenerate inputs, there is no explosion for non-degenerate inputs. Traces are averaged over 10 independent runs.

## 3.3 AVOIDING GRADIENT EXPLOSION IN DEPTH

So far, we have proven that the constructed network maintains the orthogonalization property of BN. Now, we turn our focus to the gradient analysis. The next theorem proves that the constructed network does not suffer from gradient explosion in depth for non-degenerate input matrices.

**Theorem 5.** *Let the loss function* $\mathcal{L} : \mathbb{R}^{d \times d} \to \mathbb{R}$ *be* $\mathcal{O}(1)$-*Lipschitz, and input batch* $X_0$ *be non-degenerate. Then, there exists an absolute constant* $C$ *such that for all* $\ell \leq L$ *it holds*

$$\mathbb{E}\left[\log \|\nabla_{W_\ell}\mathcal{L}(X_\ell)\|\right] \leq Cd^5(\phi(X_0)^3 + 1) \tag{10}$$

*where the expectation is over the random orthogonal weight matrices.*

**Remark 1.** *For degenerate inputs* $\phi(X_0) = \infty$ *holds, in that the bound becomes vacuous.*

**Remark 2.** *The* $\mathcal{O}(1)$-*Lipschitz condition holds in many practical settings. For example, in a classification setting, MSE and cross entropy losses obey the* $\mathcal{O}(1)$-*Lipschitz condition (see Lemma D.2).*

Note that the bound is stated for the expected value of log-norm of the gradients, which can be interpreted as bits of precision needed to store the gradient matrices. Thus, the fact that depth does not appear in any form in the upper bound of Theorem 5 points out that training arbitrarily deep MLPs with orthogonal weights will not face numerical issues that arise with Gaussian weights (Yang et al., 2019) as long as the inputs are non-degenerate. Such guarantees are necessary to ensure backpropagation will not face numerical issues.

Theorem 5 states that as long as the input samples are not linearly dependent, the gradients remain bounded for any arbitrary depth $L$. As discussed in the previous section and evidenced in Figure 2, this is necessary to avoid gradient explosion. Therefore, the upper bound provided in Theorem 5 is tight in terms of inputs constraints. Furthermore, as mentioned before, random batches sampled from commonly used benchmarks, such as CIFAR10, CIFAR100, MNIST, and FashionMNIST, are non-degenerate in most practical cases (see Section E for more details). Thus, the assumptions and thereby assertions of the theorem are valid for all practical purposes.

To the best of our knowledge, Theorem 5 is the first non-asymptotic gradient analysis that holds for networks with batch normalization and finite width. Previous results heavily rely on mean field analyses in asymptotic regimes, where the network width tends to infinity (Yang et al., 2019). While mean-field analyses have brought many insights about the rate of gradient explosion, they are often specific to Gaussian weights. Here, we show that non-Gaussian weights can avoid gradient explosion, which has previously been considered "unavoidable" (Yang et al., 2019). Figure 3 illustrates this pronounced discrepancy.

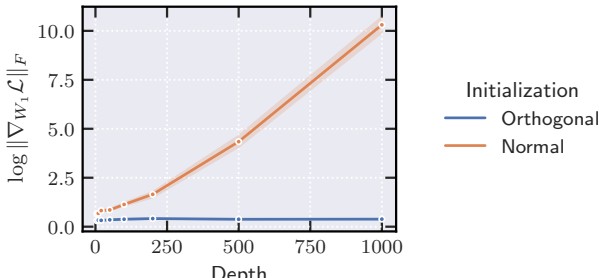

Figure 3: Logarithmic plot for the gradient norm of the first layer for networks with different number of layers evaluated on CIFAR10. For Gaussian weights (orange) the gradient-norm grows at an exponential rate, as predicted by Yang et al. (2019, Theorem 3.9), while for orthogonal weights (blue) gradients remain bounded by a constant, validating Theorem 5. Traces are averaged over 10 runs and shaded regions denote the 95% confidence intervals.

*Proof idea of Theorem 5.* The first important observation is that, due to the chain rule, we can bound the log-norm of the gradient of a composition of functions, by bounding the summation of the log-norms of the input-output Jacobian of each layer, plus two additional terms corresponding to the loss term and the gradient of the first layer in the chain. If we discount the effect of the first and last terms, the bulk of the analysis is dedicated to bounding the total sum of log-norms of per layer input-output Jacobian, i.e., the fully connected and batch normalization layers. The second observation is that because the weights are only rotations, their Jacobian has eigenvalues equal to 1. Thus, the log-norm of gradients corresponding to fully connected layers vanish. What remains is to show that for any arbitrary depth $\ell$, the log-norm of gradients of batch normalization layers also remains bounded. The main technical novelty for proving this step is showing that the log-norm of the gradient of BN layers is upper bounded by the isometry gap of pre-normalization matrices. Thus, we can invoke the exponential decay in isometry gap stated in Theorem 1 to establish a bound on the log-norm of the gradient of these layers. Finally, since the decay in isometry gap is exponentially fast, the bound on the total sum of log-norm of the gradients amounts to a geometric sum that remains bounded for any arbitrary depth $\ell$.

## 4 IMPLICATIONS ON TRAINING

In this section, we experimentally validate the benefits of avoiding gradient explosion and rank collapse for training. Thus far, we have proved that our constructed neural network with BN does not suffer from gradient explosion in Theorem 5, and does not have the rank collapse issue in depth via the orthogonalization property established in Theorem 1. We find that the constructed MLP is therefore less prone to numerical issues that arise when training deep networks.

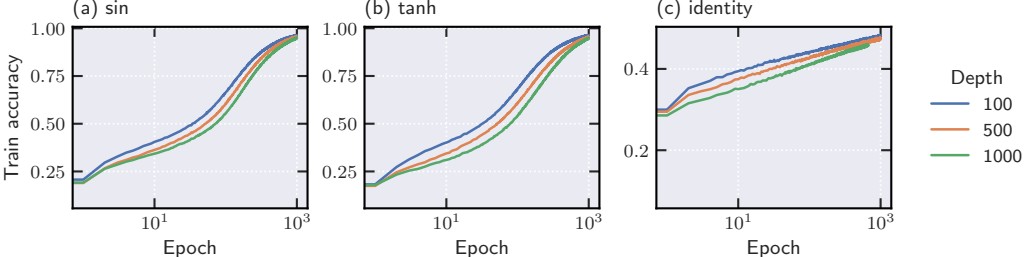

Figure 4: Contrasting the training accuracy of MLPs with BN and shaped sin, shaped tanh and identity activations, on the CIFAR10 dataset. The identity activation performs much worse than the nonlinearities, confirming that the sin and tanh networks are not operating in the linear regime. The networks are trained with vanilla SGD and the hyperparameters are width 100, batch size 100, learning rate 0.001.

By avoiding gradient explosion and rank collapse in depth, we observe that the optimization with vanilla minibatch Stochastic Gradient Descent (SGD) exhibits an almost depth-independent convergence rate for linear MLPs. In other words, the number of iterations to get to a certain accuracy does not vary widely between networks with different depths. Figure 4 (c) shows the convergence of SGD for CIFAR10, with learning rate 0.001, for MLPs with width $d = 100$ and batch size $n = 100$. While the SGD trajectory strongly diverges from the initial conditions that we analyze theoretically, Figure 4 shows that the gradients remain stable during training, as well as the fact that different depths exhibit largely similar accuracy curves.

While the empirical evidence for our MLP with linear activation is encouraging, non-linear activations are essential parts of feature learning (Nair & Hinton, 2010; Klambauer et al., 2017; Hendrycks & Gimpel, 2016; Maas et al., 2013). However, introducing non-linearity violates one of the key parts of our theory, in that it prevents representations from reaching perfect isometry (see Figure H5 in the Appendix for details on the connection between non-linearities and gradient explosion in depth). Intuitively, this is due to the fact that non-linear layers, as opposed to rotations and batch normalization, perturb the isometry of representations and prevent them from reaching zero isometry gap in depth. This problem turns out to be not just a theoretical nuisance, but to play a direct role in the gradient explosion behavior. While the situation may seem futile at first, it turns out that activation shaping (Li et al., 2022; Zhang et al., 2022; Martens et al., 2021; He et al., 2023; Noci et al., 2023) can alleviate this problem, which is discussed next. For the remainder of this section, we focus on the training of MLPs with non-linear activations, as well as standard batch normalization and fully connected layers.

## 5 ACTIVATION SHAPING BASED ON THE THEORETICAL ANALYSIS

In recent years, several works have attempted to overcome the challenges of training very deep networks by parameterizing activation functions. In a seminal work, Martens et al. (2021) propose *deep kernel shaping*, which is aimed at facilitating the training of deep networks without relying on skip connections or normalization layers, and was later extended to LeakyReLU in *tailored activation transformations* (Zhang et al., 2022). In a similar direction, Li et al. (2022) propose *activation shaping* in order to avoid a degenerate output covariance. While the mechanism proposed by Li et al. (2022) covers both smooth and non-smooth activations, we focus on their result for LeakyReLU, which consists of shaping the negative slope of the activation towards identity to ensure that the output covariance matrix remains non-degenerate when the networks becomes very deep.

Since kernel and activation shaping aim to replace normalization, they have not been used in conjunction with normalization layers. In fact, in networks with batch normalization, even linear activations have non-degenerate outputs (Daneshmand et al., 2021; Yang et al., 2019) and exploding gradients (Yang et al., 2019). Thus, shaping activations towards identity in the presence of normalization layers may seem fruitless. Remarkably, we empirically demonstrate that we can leverage activation shaping to avoid gradient explosion in depth by using a pre-activation gain at each layer.

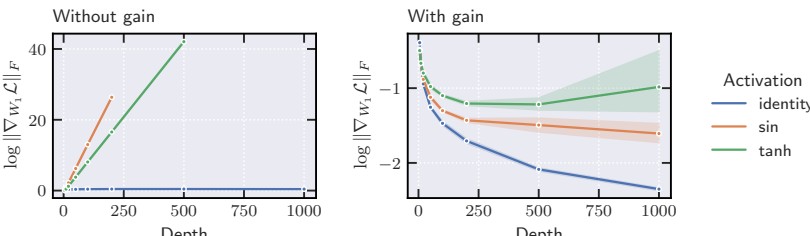

Figure 5: Logarithmic plot contrasting the effect of gain on the gradient at initialization of the first layer, for networks with different number of layers initialized with orthogonal weights, BN and different activations, evaluated on CIFAR10. The networks have hyperparameters width 100, batch size 100. Traces are averaged over 10 independent runs, with the shades showing the 95% confidence interval.

Inspired by our theory, we develop a novel activation shaping scheme for networks with BN. The main strategy consists of shaping the activation function towards a linear function across the layers.

Our activation shaping consists of tuning the gain of the activation, i.e., tuning $\alpha$ for $\sigma(\alpha x)$. We consider non-linear activations $\sigma \in \{\tanh, \sin\}$.

The special property that both $\tanh$ and $\sin$ activations have in common is that they are centered, $\sigma(0) = 0$, are differentiable around the origin $\sigma'(0) = 1$, and have bounded gradients $\sigma'(x) \leq 1, \forall x$. Therefore, by tuning the per-layer pre-activation gain $\alpha_\ell$ towards 0, the non-linearities behave akin to the identity function. This observation inspires us to study the relationship between the rate of gradient explosion for each layer as a function of the gain parameter $\alpha_\ell$. Formally, we consider an MLP with shaped activations using gain $\alpha_\ell$ for the $\ell$th layer, that has the update rule

$$X_{\ell+1} = \sigma(\alpha_\ell \text{BN}(W_\ell X_\ell)) . \tag{11}$$

Since the gradient norm has an exponential growth in depth, as shown in Figure 5, we can compute the slope of the linear growth rate of log-norm of gradients in depth. We define the rate of explosion for a model of depth $L$ and gain $\alpha_\ell$ at layer $\ell$ as the slope of the log norm of the gradients $R(\ell, \alpha_\ell)$. We show in Figure 5 that by tuning the gain properly, we are able to reduce the exponential rate of the log-norm of the gradients by diminishing the slope of the rate curve and achieve networks trainable at arbitrary depths, while still maintaining the benefits of the non-linear activation. The main idea for our activation shaping strategy is to have a bounded total sum of rates across layers, by ensuring faster decay than a harmonic series (see App. F for more details on activation shaping). Figure 5 illustrates that this activation shaping strategy effectively avoids gradient explosion while maintaining the signal propagation and orthogonality of the outputs in depth. Furthermore, Figure 4 shows that the training accuracy remains largely depth-independent. For further experiments using activation shaping, see Appendix H.

## 6    DISCUSSION

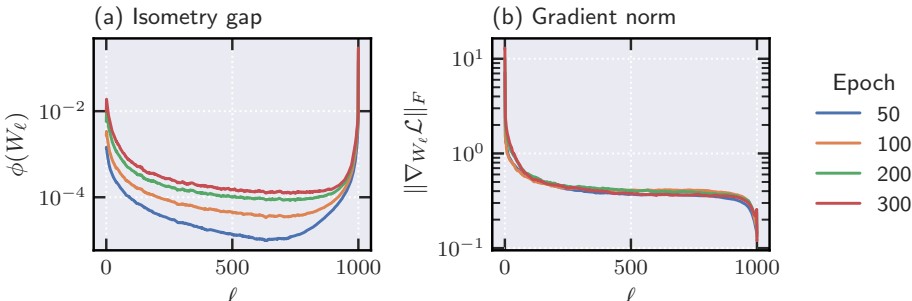

Figure 6: **Implicit orthogonality bias of SGD.** Training an MLP with width $d = 100$, batch size $n = 100$, and depth $L = 1000$, activation $\tanh$, using SGD with lr $= 0.001$ (a) Isometry gap (y-axis; log-scale) of weight matrices across all layers throughout training. (b) Gradient norms at each layer during training.

**Implicit bias of SGD towards orthogonality in optimization.**    Optimization over orthogonal matrices has been an effective approach for training deep neural networks. Enforcing orthogonality during training ensures that the spectrum of the weight matrices remains bounded, which prevents gradient vanishing and explosion in depth. Vorontsov et al. (2017) study how different orthogonality constraints affect training performance in RNNs. For example Lezcano-Casado & Martınez-Rubio (2019) leverage the exponential map on the orthogonal group, Jose et al. (2018) decompose RNN transition matrices in Kronecker factors and impose soft constraints on each factor and Mhammedi et al. (2017) introduce a constraint based on Householder matrices.

While these studies *enforce* orthogonality constraints, one of our most striking empirical observations is that when our MLP grows very deep, the middle layers remain almost orthogonal even after many steps of SGD. As shown in Figure 6, for 1000 layer networks, the middle layers remain orthogonal during training. One could hypothesize that this is due to small gradients in these layers. In Figure 6, we observe that the gradients of these middle layers are not negligible. Thus, in our MLP construction, both with linear activation and with activation shaping, the gradient dynamics have an *implicit bias* to optimize over the space of orthogonal matrices. The mechanisms underlying this implicit orthogonality bias will be an ample direction for future research.

ACKNOWLEDGMENTS & CONTRIBUTIONS

AM: proofs of Section B, driving experiments, and writing the paper. AJ: proofs of Section C and Section D, designing the activation shaping scheme, and writing the paper. FO: proposing the idea of using orthogonal weights to achieve perfect isometry, reading the proofs, help with writing. AI: reading the proofs, help with experiments and paper writing. GR: help with experimental designs for activation shaping and paper writing. HD: proposing the idea and leading the proofs, proposed using orthogonal weights to avoid gradients explosion, writing the initial draft.

AJ is funded through Swiss National Science Foundation Project Grant #200550 to Andre Kahles. HD acknowledges support from the NSF TRIPODS program (award DMS-2022448). AI acknowledges support from the Max Planck ETH Center for Learning Systems. AJ and AI were partially funded by ETH Core funding (to G.R.).

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

TABLE OF CONTENTS

The appendix is structured in the following sections:

## A  SUPPLEMENTAL RELATED WORK

Due to space constraints, we expand upon the related works in this section.

**Mean field theory for random neural networks.**  The existing analyses for random networks often rely on mean-field regimes where the network width tends to infinity (Pennington et al., 2017; Yang et al., 2019; Li et al., 2022; Pennington & Worah, 2017). However, there is a discrepancy between mean-field regimes and the practical regime of finite width. While some analyses attempt to bridge this gap (Joudaki et al., 2023a; Daneshmand et al., 2021), their results rely on technical assumptions that are hard to validate. In contrast, our non-asymptotic results hold for standard neural networks used in practice. Namely, our main assumption for avoiding rank collapse and gradient explosion is that samples in the input batch are not linearly dependent, which we will show is necessary. To go beyond mean-field regimes, we leverage recent theoretical advancements in Weingarten calculus (Weingarten, 1978; Collins, 2003; Banica et al., 2011; Collins & Śniady, 2006; Collins et al., 2022).

## B  CONDITIONAL ORTHOGONALIZATION

**Isometry after rotation.**  Our analysis is based on  (Joudaki et al., 2023b, Corollary 3), which we restate in the following Lemma:

**Lemma B.1.** *For all non-degenerate matrices $X \in \mathbb{R}^{d \times d}$, we have:*

$$\mathcal{I}(\text{BN}(X)) \geq \left(1 + \frac{variance\{\|X_{j\cdot}\|\}_{j=1}^d}{(mean\{\|X_{j\cdot}\|\}_{j=1}^d)^2}\right) \mathcal{I}(X) . \tag{12}$$

Lemma B.1 proves isometry bias of BN. The next lemma proves that isometry does not change under rotation

**Lemma B.2** (Isometry after rotation)**.** *Let $X \in \mathbb{R}^{d \times d}$ and $W \sim \mathbb{O}_d$ be a random orthogonal matrix and $X' = WX$. Then,*

$$\mathcal{I}(\text{BN}(X')) \geq \left(1 + \frac{variance\{\|X'_{j\cdot}\|\}_{j=1}^d}{(mean\{\|X'_{j\cdot}\|\}_{j=1}^d)^2}\right) \mathcal{I}(X) . \tag{13}$$

*Proof.* Using properties of the determinant, we have

$$\det(X'X'^\top) = \det(W)^2 \det(XX^\top) = \det(XX^\top), \tag{14}$$

where the last equation holds since $W$ is an orthogonal matrix. Furthermore,

$$\mathrm{Tr}(X'X'^\top) = \mathrm{Tr}(WXX^\top W^\top) \tag{15}$$

$$= \mathrm{Tr}(XX^\top \underbrace{W^\top W}_{=I}) \tag{16}$$

$$= \mathrm{Tr}(XX^\top). \tag{17}$$

Combining the last two equations with Lemma B.1 concludes the proof. □

**Increasing isometry with rotations and** BN. The last lemma proves the isometry bias does not decrease with rotation and BN. However, this does not prove a strict decrease in isometry with BN and rotation. The next lemma proves there exists an orthogonal matrix for which the isometry is strictly increasing.

**Corollary B.3** (Increasing isometry). *Let $X \in \mathbb{R}^{d \times d}$ and denote its singular value decomposition $X = U \mathrm{diag}\left(\{\sigma_i\}_{i=1}^d\right) V^\top$, where $U$ and $V$ are orthogonal matrices. Then, we have:*

$$\mathcal{I}(\mathrm{BN}(U^\top H)) = 1. \tag{18}$$

*Proof.* Let $S = \mathrm{diag}\left(\{\sigma_i\}_{i=1}^d\right)$ be the diagonal matrix containing the singular values of $X$. Then, we have:

$$\mathrm{BN}(U^\top X) = \mathrm{BN}(U^\top U S V^\top) \tag{19}$$

$$= \mathrm{BN}(SV^\top) \tag{20}$$

$$= \mathrm{diag}(SV^\top V S)^{-\frac{1}{2}} S V^\top \tag{21}$$

$$= S^{-1} S V^\top \tag{22}$$

$$= V^\top, \tag{23}$$

which has maximum isometry 1 since it's an orthonormal matrix. □

Thus, there exists a rotation that increases isometry with BN for each non-orthogonal matrix. The proof of the last corollary is based on a straightforward application of Lemma 2.

**Orthogonalization with randomness.** The isometric is non-decreasing in Lemma 2 and provably increases for a certain rotation matrix (as stated in the last corollary). Hence, it is possible to increase isometry with random orthogonal matrices.

**Theorem B.4.** *Suppose $W \sim \mathbb{O}_d$ is a matrix drawn from $\mathbb{O}_d$ such that $W \overset{d}{=} WU$ for all orthogonal matrices $U$. Let $\{\lambda_i\}_{i=1}^d$ be the eigenvalues of $XX^\top$. Then,*

$$\mathbb{E}_W\left[\mathcal{I}(\mathrm{BN}(WX))|\,X\right] \geq \left(\frac{1}{1 - \frac{\sum_{k=1}^d (\lambda_k - 1)^2}{2d^2(d+2)}}\right) \mathcal{I}(X) \tag{24}$$

*holds for all $X = \mathrm{BN}(\cdot)$, with equality for orthogonal matrices.*

**Remark 3.** *Note that the assumption on $X = \mathrm{BN}(\cdot)$, can be viewed as an induction hypothesis, in that we can recursively apply this theorem to arrive at a quantitative rate at depth.*

Notably, $\sum_{i=1}^d \lambda_i = d$ if $X = \mathrm{BN}(\cdot)$. Hence, one can expect that $\sum_{i=1}^d \lambda_i^2 < d^2$ for all full random matrices $X$ in form of $X = \mathrm{BN}(\cdot)$.

*Proof.* We need to compute the variance/mean ratio in Lemma 2. Let $X \in \mathbb{R}^{d \times d}$ have SVD decomposition $X = U \mathrm{diag}\{\sigma_i\}_{i=1}^d V^\top$ where $U$ and $V$ are orthogonal matrix and $\sigma_i^2 = \lambda_i$. Since

the distribution of $W$ is invariant to transformations with orthogonal matrices, the distribution of $W$ equates those of $X' = W\text{diag}\{\sigma_i\}V^\top$. It easy to check that

$$\|X'_{j\cdot}\| = \sqrt{\sum_{i=1}^{d}\sigma_i^2 W_{ji}^2} = \sqrt{\sum_{i=1}^{d}\lambda_i W_{ji}^2}\,. \tag{25}$$

Thus,

$$\sum_{j=1}^{d}\|X'_{j\cdot}\|^2 = \sum_{i=1}^{d}\lambda_i = d, \tag{26}$$

where the last equality holds due to the batch normalization. Thus, we have

$$\mathbb{E}\left[\frac{d\sum_{j=1}^{d}\|X'_{j\cdot}\|^2}{(\sum_{j=1}^{d}\|X'_{j\cdot}\|)^2}\right] = \mathbb{E}\left[\frac{d^2}{(\sum_{j}\|X'_{j\cdot}\|)^2}\right] \geq \frac{d^2}{\mathbb{E}\left[\left(\sum_{j=1}^{d}\|X'_{j\cdot}\|\right)^2\right]}\,. \tag{27}$$

We need to estimate

$$\mathbb{E}\left[\|X'_i\|\|X'_j\|\right] = \mathbb{E}\left[\left(\sum_{k=1}^{d}\lambda_k W_{ik}^2\right)^{\frac{1}{2}}\left(\sum_{k=1}^{d}\lambda_k W_{jk}^2\right)^{\frac{1}{2}}\right]\,. \tag{28}$$

Since square root function is concave, we have $\sqrt{x} \leq 1 + \frac{1}{2}(x-1)$. Thus

$$\mathbb{E}\left[\|X'_i\|\|X'_j\|\right] \leq \mathbb{E}\left[\left(1 + 0.5\sum_{k=1}^{d}(\lambda_k - 1)W_{ik}^2\right)\left(1 + 0.5\sum_{k=1}^{d}(\lambda_k - 1)W_{jk}^2\right)\right] \tag{29}$$

$$= 1 + \frac{1}{4}\sum_{k,q}(\lambda_k - 1)(\lambda_q - 1)\mathbb{E}\left[W_{ik}^2 W_{jq}^2\right] \tag{30}$$

$$= 1 + \frac{1}{4}\left[\sum_{k\neq q}(\lambda_k - 1)(\lambda_q - 1)\underbrace{\mathbb{E}\left[W_{ik}^2 W_{jq}^2\right]}_{E_1} + \sum_{k=q}(\lambda_k - 1)(\lambda_k - 1)\underbrace{\mathbb{E}\left[W_{ik}^2 W_{jk}^2\right]}_{E_2}\right], \tag{31}$$

where in the first equality we have used the fact that the cross terms reduce, where the expectations are applications of Weingarten calculus:

$$\mathbb{E}\left[0.5\sum_{k=1}^{d}(\lambda_k - 1)W_{ik}^2 + 0.5\sum_{k=1}^{d}(\lambda_k - 1)W_{jk}^2\right] = \mathbb{E}\left[0.5\sum_{k=1}^{d}\lambda_k W_{ik}^2 + 0.5\sum_{k=1}^{d}\lambda_k W_{jk}^2 - 1\right] \tag{32}$$

$$= 0.5\sum_{k=1}^{d}\lambda_k\mathbb{E}\left[W_{ik}^2\right] + 0.5\sum_{k=1}^{d}\lambda_k\mathbb{E}\left[W_{jk}^2\right] - 1 \tag{33}$$

$$= \frac{0.5}{d}\sum_{k=1}^{d}\lambda_k + \frac{0.5}{d}\sum_{k=1}^{d}\lambda_k - 1 \tag{34}$$

$$= 0\,. \tag{35}$$

The main quantity we must compute is an expectation of polynomials taken over the Haar measure of the Orthogonal group $O(n)$. To carry out the computation, we make use of Weingarten calculus (Banica et al., 2011; Collins & Śniady, 2006; Weingarten, 1978). More specifically, we make use of the of the Weingarten formula, studied by (Collins & Śniady, 2006; Collins et al., 2022):

$$\int_{O(n)} r_{i_1 j_1} r_{i_2 j_2}\ldots r_{i_{2d} j_{2d}}\, d\mu(O(n)) = \sum_{\sigma\in\mathcal{M}_{2d}}\sum_{\sigma\in\mathcal{M}_{2n}}\Delta_\sigma(\boldsymbol{i})\Delta_\sigma(\boldsymbol{j})\text{Wg}^O(\sigma^{-1}\tau), \tag{36}$$

where $\mu(O(n))$ is the Haar measure of Orthogonal group. For an in depth explanation of each quantity in the Weingarten formula, we refer the reader to Collins et al. (2022, Section 5.2).

The quantity we focus on is $\mathbb{E}_W\left[W_{ik}W_{ik}W_{jq}W_{jq}\right]$. We will do the computation on multiple cases, based on the equalities of $k, q$. Notice that $i \neq j$ in all cases. It suffices if we focus on the two distinct cases: $E_1 = \mathbb{E}_W\left[W_{ik}^2 W_{jq}^2\right]$ ($k \neq q$) and $E_2 = \mathbb{E}_W\left[W_{ik}^2 W_{jk}^2\right]$ ($k = q$).

We first compute $E_1$.

Following the procedure from Collins et al. (2022, Section 5.2), we take the index sequences to be $\boldsymbol{i} = (i, i, j, j)$ and $\boldsymbol{j} = (k, k, q, q)$. Similarly, we get $\Delta_\sigma(\boldsymbol{i}) = \Delta_\tau(\boldsymbol{j}) = 1$ only if $\sigma = \{\{1, 2\}, \{3, 4\}\}$ and $\tau = \{\{1, 2\}, \{3, 4\}\}$.

Consdering $\sigma, \tau$ as permutations, we get:

$$\sigma = \begin{pmatrix} 1 & 2 & 3 & 4 \\ 1 & 2 & 3 & 4 \end{pmatrix}, \tag{37}$$

$$\tau = \begin{pmatrix} 1 & 2 & 3 & 4 \\ 1 & 2 & 3 & 4 \end{pmatrix}, \tag{38}$$

$$\sigma^{-1}\tau = \begin{pmatrix} 1 & 2 & 3 & 4 \\ 1 & 2 & 3 & 4 \end{pmatrix}, \tag{39}$$

where $\sigma^{-1}\tau$ has coset-type $[1, 1]$. Finally, we plug the results back into the formula and we obtain:

$$E_1 = \mathbb{E}_W\left[W_{ik}^2 W_{jq}^2\right] = \mathrm{Wg}^O([1, 1]) = \frac{d+1}{d(d+2)(d-1)},$$

where the last equality is based on the results in Section 7 of (Collins & Matsumoto, 2009).

We compute $E_2$.

Similar to the previous expression, we take the index sequences to be to be $\boldsymbol{i} = (i, i, j, j)$ and $\boldsymbol{j} = (k, k, k, k)$. Thus, we obtain $\Delta_\sigma(\boldsymbol{i}) = \Delta_\tau(\boldsymbol{j}) = 1$ only if $\sigma = \{\{1, 2\}, \{3, 4\}\}$ and $\tau_1 = \{\{1, 2\}, \{3, 4\}\}$, $\tau_2 = \{\{1, 3\}, \{2, 4\}\}$, $\tau_1 = \{\{1, 4\}, \{2, 3\}\}$. Similarly, we compute $\sigma^{-1}\tau_i$ for $i \in \{1, 2, 3\}$. Notice that $\sigma$ is the identity permutation, thus yielding $\sigma^{-1}\tau_i = \tau_i$, with the coset-types $[1, 1], [2], [2]$ respectively, for each $i \in \{1, 2, 3\}$.

Plugging back into the original equation, we obtain:

$$E_2 = \mathbb{E}_W\left[W_{ik}^2 W_{jk}^2\right] = \mathrm{Wg}^O([1, 1]) + 2\mathrm{Wg}^O([2]) \tag{40}$$

$$= \frac{d+1}{d(d+2)(d-1)} + 2\frac{-1}{d(d+2)(d-1)} \tag{41}$$

$$= \frac{d-1}{d(d+2)(d-1)} . \tag{42}$$

Thus, plugging back into the original inequality, we obtain:

$$\mathbb{E}\|X_i'\|\|X_j'\| \leq 1 + \frac{1}{4}\left[\sum_{k \neq q}(\lambda_k - 1)(\lambda_q - 1)\frac{d+1}{d(d+2)(d-1)} + \sum_{k=q}(\lambda_k - 1)(\lambda_k - 1)\frac{d-1}{d(d+2)(d-1)}\right] \tag{43}$$

$$= 1 + \frac{1}{4d(d+2)(d-1)}\left[\underbrace{\sum_{k \neq q}(\lambda_k - 1)(\lambda_q - 1)}_{S_{\neq}} - \underbrace{\sum_{k=q}(\lambda_k - 1)(\lambda_k - 1)}_{S_=}\right] \tag{44}$$

$$= 1 - \frac{\sum_k(\lambda_k - 1)^2}{2d(d+2)(d-1)}, \tag{45}$$

where we have used that $S_{\neq} + S_{=} = \sum_{k,q}(\lambda_k - 1)(\lambda_q - 1) = (\sum_{k=1}^{d}(\lambda_k - 1))^2 = 0$ in the last equality.

Thus, we obtain:

$$\mathbb{E}\left[\left(\sum_j \|X'_{j.}\|\right)^2\right] = \mathbb{E}\left[\sum_j \|X'_{j.}\|^2 + 2\sum_{i<j} \|X'_{i.}\|\|X'_{j.}\|\right] \tag{46}$$

$$= d + 2\sum_{i<j} \mathbb{E}\left[\|X'_{i.}\|\|X'_{j.}\|\right] \tag{47}$$

$$\leq d + (d^2 - d)\left(1 - \frac{\sum_k(\lambda_k - 1)^2}{2d(d+2)(d-1)}\right) \tag{48}$$

$$= d^2 - \frac{\sum_k(\lambda_k - 1)^2}{2(d+2)} . \tag{49}$$

$\square$

**Corollary B.5** (Isometry gap bound). *Suppose the same setup as in Theorem 3. Then, we have:*

$$\mathbb{E}_W[\phi(X')|X] \leq \phi(X) + \log\left(1 - \frac{\sum_k(\lambda_k - 1)^2}{2d^2(d+2)}\right). \tag{50}$$

**Remark 4.** *Notice that the term $\frac{\sum_{k=1}^{d}(\lambda_k-1)^2}{2d^2(d+2)} = \mathcal{O}(\frac{1}{d})$, yielding $\log\left[1 - \frac{\sum_{k=1}^{d}(\lambda_k-1)^2}{2d^2(d+2)}\right] \leq 0$.*

*Proof.* From Lemma 2, we know that:

$$-\log \mathcal{I}(\text{BN}(X')) \leq -\log \mathcal{I}(X) - \log \frac{d^2}{\left(\sum_{j=1}^{d} \|X'_{j.}\|\right)^2} \tag{51}$$

$$\implies -\log \mathcal{I}(\text{BN}(X')) \leq -\log \mathcal{I}(X) - \log d^2 + \log\left(\sum_{j=1}^{d} \|X'_{j.}\|\right)^2 \tag{52}$$

$$\implies \mathbb{E}_W[-\log \mathcal{I}(\text{BN}(X'))|X] \leq -\log \mathcal{I}(X) - \log d^2 + \mathbb{E}_W\left[\log\left(\sum_{j=1}^{d} \|X'_{j.}\|\right)^2 \Big| X\right] \tag{53}$$

$$\leq -\log \mathcal{I}(X) - \log d^2 + \log \mathbb{E}_W\left[\left(\sum_{j=1}^{d} \|X'_{j.}\|\right)^2 \Big| X\right] \tag{54}$$

$$\leq -\log \mathcal{I}(X) - \log d^2 + \log\left(d^2 - \frac{\sum_k(\lambda_k - 1)^2}{2(d+2)}\right) \tag{55}$$

$$\leq -\log \mathcal{I}(X) + \log\left(1 - \frac{\sum_k(\lambda_k - 1)^2}{2d^2(d+2)}\right), \tag{56}$$

where in inequality 54 we have used the fact that $\mathbb{E}[\log X] \leq \log \mathbb{E}[X]$ and in inequality 55 we have used the bound obtained in proof Theorem 3, equation 49. Thus, we obtain:

$$\mathbb{E}_W[\phi(X')|X] \leq \phi(X) + \log\left(1 - \frac{\sum_k(\lambda_k - 1)^2}{2d^2(d+2)}\right) . \tag{57}$$

$\square$

## C  ISOMETRY GAP DECAY RATE

Before we start with the main part of our analysis, let us establish a simple result on the relation between isometry gap and orthogonality:

**Lemma C.1** (Isometry gap and orthogonality). *If $\phi(X) \leq \frac{c}{16d}$, then eigenvalues of $X^\top X$ are within $[1 - c, 1 + c]$.*

Note that, in order to simplify the calculations, we use the fact that $\frac{1}{d(d+2)} \approx \frac{1}{d^2}$ in the following proofs.

Based on the conditional expectation in Corollary B.5, we have:

$$\mathbb{E}\left[\phi(X^{\ell+1})|X^\ell\right] \leq \phi(X^\ell) - \frac{\sigma_\lambda(X^\ell)}{2d^2} . \tag{58}$$

Now, we prove a lemma that is conditioned on the previous layer isometry gap being smaller or larger than $\frac{1}{16d}$.

**Lemma C.2** (Isometry gap conditional bound). *For $X^\ell$ being the representations of an MLP under our setting, we have:*

$$\mathbb{E}\left[\phi(X^{\ell+1})\Big|X_\ell, \phi(X^\ell) \leq \frac{1}{16d}\right] \leq \phi(X^\ell)\left(1 - \frac{1}{2d^2}\right), \tag{59}$$

$$\mathbb{E}\left[\phi(X^{\ell+1})\Big|X_\ell, \phi(X^\ell) > \frac{1}{16d}\right] \leq \phi(X^\ell) - \frac{1}{32d^3} . \tag{60}$$

*Proof of Lemma C.2.* Let $\lambda_k = 1 + \epsilon_k$, and assume without loss of generality that $\sum_{k=1}^d \epsilon_k = 0$. Then, using the numerical inequality $\log(1 + x) \geq x - x^2$, when $|x| \leq \frac{1}{2}$ we have:

$$\sigma_\lambda(X) = \frac{1}{d}\sum_{k=1}^d \epsilon_k^2 \tag{61}$$

$$\max_k |\epsilon_k| \leq \frac{1}{2} \implies \phi(X) = -\frac{1}{d}\sum_{k=1}^d \log(1 + \epsilon_k) \leq -\frac{1}{d}\sum_{k=1}^d (\epsilon_k - \epsilon_k^2) = \sigma_\lambda(X) . \tag{62}$$

Altogether, we have

$$\max_i |\epsilon_i| \leq \frac{1}{2} \implies \phi(X) \leq \sigma_\lambda(X) . \tag{63}$$

Now, we can restate the condition in terms of an inequality on the isometry gap. Thus, we can write:

$$d\phi(X) = -\sum_{i=1}^d \log \lambda_i = -\sum_{i=1}^d \log(1 + \epsilon_i) \geq -\sum_{i=1}^d \left(\epsilon_i - \frac{3\epsilon_i^2}{6 + 4\epsilon_i}\right) = \sum_{i=1}^d \frac{3\epsilon_i^2}{6 + 4\epsilon_i}, \tag{64}$$

where we used the fact that $\sum_i \lambda_i = d$ implying $\sum_i \epsilon_i = 0$ and also used the inequality $\log(1+x) \leq x - \frac{6+x}{6+4x}$ when $x \geq -1$ for $\epsilon_i$'s. Note that because $|\epsilon_i| \leq \frac{1}{2}$, we get $6 + 4\epsilon_i \geq 4$, all terms on the right-hand side are positive, implying that each term is bounded by the upper bound:

$$\frac{3\epsilon_i^2}{6 + 4\epsilon_i} \leq d\phi(X) \quad \forall i \in \{1, 2, \dots, d\} . \tag{65}$$

By construction, we have $\epsilon_i \geq -1$ and $\frac{3\epsilon_i^2}{6+4\epsilon_i} \leq \frac{1}{16}$. Since $6 + 4\epsilon_i \geq 2$, we can multiply both sides by $6 + 4\epsilon_i$ and conclude $3\epsilon_i^2 - \frac{6+4\epsilon_i}{16} \leq 0$. We can now solve the quadratic equation and obtain $\frac{1-\sqrt{74}}{24} \leq \epsilon_i \leq \frac{1+\sqrt{74}}{24}$ which numerically becomes $-0.35 \leq \epsilon_i \leq 0.4$, implying $|\epsilon_i| < 0.5$.

By solving the quadratic equation above we can guarantee that

$$\phi(X) \leq \frac{1}{16d} \implies \max_i |\epsilon_i| \leq \frac{1}{2} \implies \phi(X) \leq \sigma_\lambda(X) . \tag{66}$$

Furthermore, we can restate the condition on maximum using:

$$\max_k |\epsilon_k| = \sqrt{\max_k \epsilon_k^2} \le \sqrt{\sum_k \epsilon_k^2} = \sqrt{d \sigma_\lambda(X)} \tag{67}$$

and conclude that

$$\sigma_\lambda(X) \le \frac{1}{4d} \implies \max_i |\epsilon_i| \le \frac{1}{2} \implies \phi(X) \le \sigma_\lambda(X). \tag{68}$$

Using this statement, we have

$$\sigma_\lambda(X) \le \frac{1}{16d} \implies \phi(X) \le \sigma_\lambda(X) \implies \phi(X) \le \frac{1}{16d}. \tag{69}$$

If we negate and flip the two sides we arrive at

$$\phi(X) > \frac{1}{16d} \implies \sigma_\lambda(X) > \frac{1}{16d}. \tag{70}$$

Thus, we can simplify the recurrence

$$\mathbb{E}[\phi(X^{\ell+1})|X^\ell] \le \phi(X^\ell) - \frac{\sigma_\lambda(X^\ell)}{2d^2} \tag{71}$$

as follows

$$\mathbb{E}\left[\phi(X^{\ell+1})\middle|X^\ell, \phi(X^\ell) \le \frac{1}{16d}\right] \le \phi(X^\ell)\left(1 - \frac{1}{2d^2}\right), \tag{72}$$

$$\mathbb{E}\left[\phi(X^{\ell+1})\middle|X^\ell, \phi(X^\ell) > \frac{1}{16d}\right] \le \phi(X^\ell) - \frac{1}{32d^3}, \tag{73}$$

where we used equation 66 in the first one and equation 70 in the second one. $\qquad\square$

*Proof of Theorem 1.* From Lemma B.1, we know that $\phi(X^0) \ge \phi(X^1) \ge \cdots \ge \phi(X^L) \ge 0$, for any layer $0 \le \ell \le L$. Thus, we get using equation 60:

$$\mathbb{E}\left[\phi(X^{\ell+1})\middle|X^\ell, \phi(X^\ell) > \frac{1}{16d}\right] \le \phi(X^\ell) - \frac{1}{32d^3} \tag{74}$$

$$= \left(1 - \frac{1}{32d^3\phi(X^\ell)}\right)\phi(X^\ell) \tag{75}$$

$$\le \left(1 - \frac{1}{32d^3\phi(X^0)}\right)\phi(X^\ell), \tag{76}$$

where in the last step we have used the fact that $\phi(X^\ell) \le \phi(X^0)$.

Thus, we can combine equation 59 and equation 60 and obtain:

$$\mathbb{E}[\phi(X^{\ell+1})|X^{\ell}] = \mathbb{E}\left[\phi(X^{\ell+1})\Big|X^{\ell}, \phi(X^{\ell}) \leq \frac{1}{16d}\right]\mathbf{1}_{\phi(X^{\ell}) \leq \frac{1}{16d}} \tag{77}$$

$$+ \mathbb{E}\left[\phi(X^{\ell+1})\Big|X^{\ell}, \phi(X^{\ell}) > \frac{1}{16d}\right]\mathbf{1}_{\phi(X^{\ell}) > \frac{1}{16d}} \tag{78}$$

$$\leq \max\left(\mathbb{E}\left[\phi(X^{\ell+1})\Big|X^{\ell}, \phi(X^{\ell}) \leq \frac{1}{16d}\right], \mathbb{E}\left[\phi(X^{\ell+1})\Big|X^{\ell}, \phi(X^{\ell}) > \frac{1}{16d}\right]\right) \tag{79}$$

$$\leq \max\left(1 - \frac{1}{2d^2}, 1 - \frac{1}{32d^3\phi(X^0)}\right)\phi(X^{\ell}) \tag{80}$$

$$= \left(1 - \min\left(\frac{1}{2d^2}, \frac{1}{32d^3\phi(X^0)}\right)\right)\phi(X^{\ell}) \tag{81}$$

$$= \left(1 - \frac{1}{\max(2d^2, 32d^3\phi(X^0))}\right)\phi(X^{\ell}) \tag{82}$$

$$\leq \exp\left[-\underbrace{\frac{1}{\max(2d^2, 32d^3\phi(X^0))}}_{k}\right]\phi(X^{\ell}) \tag{83}$$

$$= \exp\left(-\frac{1}{k}\right)\phi(X^{\ell}). \tag{84}$$

By iterated expectations over $X^{\ell}$ we get:

$$\mathbb{E}[\phi(X^{\ell+1})] \leq \exp\left(-\frac{1}{k}\right)\mathbb{E}[\phi(X^{\ell})] \leq \exp\left(-\frac{\ell}{k}\right)\phi(X^0). \tag{85}$$

Note that since $\max(2d^2, 32d^3\phi(X^0)) \leq 2d^2(1 + 16d\phi(X^0))$, we can conclude the proof. $\qquad \square$

# D  GRADIENT NORM BOUND

In the following section, we denote by $H^\ell = W^\ell X^\ell$ the pre-normalization values. Moreover, we define as $\mathcal{F}_L : \mathbb{R}^{d \times d} \to \mathbb{R}^{C \times d}$, where $C$ is the number of output classes, as the functional composition of an $L$ layers MLP, following the update rule defined in equation 3, i.e.:

$$\mathcal{F}_L(X_L) = \text{BN}(W^L \mathcal{F}_{L-1}(X_{L-1})) . \tag{86}$$

Let us restate the theorem for completeness of the appendix:

**Theorem D.1** (Restated Thm. 5)**.** *For any $\mathcal{O}(1)$-Lipschitz loss function $L$ and non-degenerate input $X_0$, we have:*

$$\log \left\| \frac{\partial \mathcal{L}}{\partial W^\ell} \right\| \lesssim d^5 \left( \phi(X_0)^3 + 1 - e^{-\frac{L}{32d^4}} \right) \tag{87}$$

*holds for all $\ell \le L$, where possibly $L \to \infty$.*

In particular, the following lemma guarantees that the Lipschitz conditions are met for practical loss functions:

**Lemma D.2.** *In a classification setting, cross entropy and mean squared error losses are $\mathcal{O}(1)$-Lipschitz.*

The main idea for the feasibility of this theorem is the presence of perfectly isometric weight matrices that are orthonormal, and the linear activation that does not lead to vanishing or exploding gradients. The only remaining layers to be analyzed are the batch normalization layers. Thus, our main goal is to show that the sum of log-norm gradient of BN layers remains bounded even if the network has infinite depth. To do so, we shall relate the norm of the gradient of those layers to the isometry gap of representations, and use the bounds from the previous section to establish that the log-norm sum is bounded.

*Proof of Thm. D.1.* Now, considering an $L$ layer deep model, where $L$ can possibly be $L \to \infty$, we can finalize the proof of Theorem 5. Consider an MLP model as defined in equation 3. Let $H^L = W^L X^L$ be the logits of the model, where $H^L \in \mathbb{R}^{C \times d}$, $W^L \in \mathbb{R}^{C \times d}$, $W^L$ is an orthogonal matrix and $C$ is the number of output classes. Denote as $\mathcal{L}(H^L, y)$ the loss of model for an input matrix, with ground truth $y$. Then, applying the chain rule, we have:

$$\frac{\partial \mathcal{L}}{\partial W^\ell} = \frac{\partial \mathcal{L}}{\partial H^L} \frac{\partial H^L}{\partial X^L} \frac{\partial X^L}{\partial X^{L-1}} \cdots \frac{\partial X^{\ell+2}}{\partial X^{\ell+1}} \frac{\partial X^{\ell+1}}{\partial H^\ell} \frac{\partial H^\ell}{\partial W^\ell} . \tag{88}$$

By taking the logarithm of the norm of each factor and applying Lemma D.12, we get:

$$\log \left\| \frac{\partial \mathcal{L}}{\partial W^\ell} \right\| \le \log \left\| \frac{\partial \mathcal{L}}{\partial H^L} \right\| + \log \underbrace{\left\| \frac{\partial H^L}{\partial X^L} \right\|}_{\|W^L\|} + \sum_{k=\ell+1}^{L} \log \left\| \frac{\partial X^{k+1}}{\partial X^k} \right\| + \log \underbrace{\left\| \frac{\partial X^{\ell+1}}{\partial H^\ell} \right\|}_{\|J_{\text{BN}}(H^\ell)\|} + \log \left\| \frac{\partial H^\ell}{\partial W^\ell} \right\| \tag{89}$$

$$\le \log \left\| \frac{\partial \mathcal{L}}{\partial H^L} \right\| + \sum_{k=\ell}^{L} \log \left\| J_{\text{BN}}(H^k) \right\| + \log \left\| \frac{\partial H^\ell}{\partial W^\ell} \right\| . \tag{90}$$

where $\log \underbrace{\left\| \frac{\partial H^L}{\partial X^L} \right\|}_{\|W^L\|} = \log \left\| W^L \right\| = 0$, since the orthogonal matrix $W^L$ has operator norm 1.

Since $\frac{\partial H^\ell}{\partial W^\ell} = X^\ell$ and $X^\ell = \text{BN}(H^{\ell-1})$ is batch normalized, this means that $\left\| \frac{\partial H^\ell}{\partial W^\ell} \right\| \le d$. Thus, the main part is to bound the Jacobian log-norms, which is provided by the following lemma:

**Lemma D.3.** *We have*

$$\sum_{k=1}^{L} \log \left\| J_{\text{BN}}(H^k) \right\| \lesssim d^5 (\phi(X_0)^3 + 1 - e^{-\frac{L}{32d^4}}) . \tag{91}$$

Finally, we can plug the bound from Lemma D.3 in equation 90 and obtain the conclusion:

$$\log\left\|\frac{\partial\mathcal{L}}{\partial W^\ell}\right\| \lesssim \log\left\|\frac{\partial\mathcal{L}}{\partial H^L}\right\| + \log d + d^5\left(\phi(X_0)^3 + 1 - e^{-\frac{L}{32d^4}}\right) . \tag{92}$$

Note that, for $L \to \infty$, we get:

$$\log\left\|\frac{\partial\mathcal{L}}{\partial W^\ell}\right\| \lesssim \log\left\|\frac{\partial\mathcal{L}}{\partial H^L}\right\| + \log d + d^5(\phi(X_0)^3 + 1) . \tag{93}$$

In order to conclude the bound, it suffices to show that the norm of the gradient of the loss with respect to the logits is bounded, which is the objective of Lemma D.2. $\qquad\square$

*Proof of Lemma D.3.* The proof of this lemma is chiefly relying on the following bound on the Jacobian of batch normalization layers, which we will state and prove beforehand.

**Lemma D.4** (Log-norm bound). *If $X \in \mathbb{R}^{d\times d}$ is the input to a BN layer, its Jacobian operator norm is bounded by*

$$\log\|J_{\text{BN}}(X)\|_{op} \leq d\phi(X) + 1 . \tag{94}$$

*Furthermore, if $\phi(X) \leq \frac{1}{16d}$, then we have*

$$\log\|J_{\text{BN}}(X)\|_{op} \leq 2\sqrt{d\phi(X)} . \tag{95}$$

Based on the lemma above, we shall define $S$ as the hitting time, corresponding to the first layer in our case, that the isometry gap drops below the critical value of $\frac{1}{16d}$:

$$S = \min\left\{\ell : \phi(X^\ell) \leq \frac{1}{16d}\right\} . \tag{96}$$

So, we first bound the total log-grad norm for layers 1 up to $S$, and subsequently $S+1$ up to $L$:

$$\log\|J_{\mathcal{F}_L}(X)\|_{op} \leq \sum_{\ell=1}^{L} \log\|J_{\text{BN}}(X^\ell)\|_{op} \tag{97}$$

$$\leq \sum_{\ell=1}^{S}(d\phi(X^\ell) + 1) + 2\sum_{\ell=S+1}^{L}\sqrt{d\phi(X^\ell)} \tag{98}$$

$$\leq \sum_{\ell=1}^{S}(d\phi(X^0) + 1) + 2\sum_{\ell=S+1}^{L}\sqrt{d\phi(X^\ell)} \tag{99}$$

$$= S(d\phi(X^0) + 1) + 2\sum_{\ell=S+1}^{L}\sqrt{d\phi(X^\ell)}, \tag{100}$$

where we have used that $\phi(X^0)$ as an upper bound on $\phi(X^\ell)$.

Thus, taking expectation we get:

$$\mathbb{E}\log\|J_{\mathcal{F}_L}(X)\|_{op} \leq (d\phi(X^0) + 1)\mathbb{E}[S] + 2\sum_{\ell=S+1}^{L}\mathbb{E}\sqrt{d\phi(X^\ell)} . \tag{101}$$

Note that $S$ is a random variable, which is why the expectation over the number of layers appears at the last line. Thus, we can bound the log-norm by bounding $\mathbb{E}[S]$ and the summation separately.

**Lemma D.5** (stopping time bound). *We have $\mathbb{E}[S] \lesssim 512d^4\phi(X^0)^2$ if $\phi(X_0) > \frac{1}{16d}$, and $\mathbb{E}[S] = 0$ if $\phi(X_0) \leq \frac{1}{16d}$.*

**Lemma D.6** (second phase bound). *We have*

$$\sum_{\ell=S+1}^{L}\mathbb{E}\sqrt{d\phi(X^\ell)} \leq 32d^{4.5}\phi(X_0)^{0.5}\left(1 - e^{-\frac{L}{32d^4}}\right) .$$

Thus, we have the following 2 cases, based on whether $\phi(X_0)$ is below or over the $\frac{1}{16d}$ threshold. If we plug the bounds in equation 101 we get the following.

If $\phi(X_0) \leq \frac{1}{16d}$, then:

$$\mathbb{E}\log\|J_{\mathcal{F}_L}(X)\|_{op} \leq 32d^{4.5}\phi(X_0)^{0.5}\left(1 - e^{-\frac{L}{32d^4}}\right) \tag{102}$$

$$\lesssim d^{4.5}\phi(X_0)^{0.5}\left(1 - e^{-\frac{L}{32d^4}}\right), \tag{103}$$

and if $\phi(X_0) > \frac{1}{16d}$ then:

$$\mathbb{E}\log\|J_{\mathcal{F}_L}(X)\|_{op} \leq 512d^4\phi(X_0)^2(1 + d\phi(X_0)) + 32d^4\left(1 - e^{-\frac{L}{32d^4}}\right) \tag{104}$$

$$\lesssim d^5\phi(X_0)^3 + d^4\left(1 - e^{-\frac{L}{32d^4}}\right) \tag{105}$$

$$\lesssim d^4\left(d\phi(X_0)^3 + 1 - e^{-\frac{L}{32d^4}}\right). \tag{106}$$

In fact, the maximum of the two bounds is

$$\mathbb{E}\log\|J_{\mathcal{F}_L}(X)\|_{op} \lesssim d^5\left(\phi(X_0)^3 + 1 - e^{-\frac{L}{32d^4}}\right). \tag{107}$$

$\square$

*Proof of Lemma D.6.* By the bound in Lemma C.2, we have

$$\mathbb{E}\left[\phi(X^{\ell+1})\middle|\phi(X^\ell) \leq \frac{1}{16d}\right] \leq \phi(X^\ell)\left(1 - \frac{1}{2d^2}\right). \tag{108}$$

Since we assumed $\ell \geq S$, the conditional inequality always holds and thus we have the Markov bound

$$\ell \geq S \implies q := \Pr\left\{\phi(X^{\ell+1}) \geq \left(1 - \frac{1}{4d^2}\right)\phi(X^\ell)\right\} \leq \frac{1 - \frac{1}{2d^2}}{1 - \frac{1}{4d^2}} \leq 1 - \frac{1}{4d^2}. \tag{109}$$

We define as failure the event $\bar{A} = \{\phi(X^{\ell+1}) \geq (1 - \frac{1}{4d^2})\phi(X^\ell)\}$ with probability $q$, and conversely as success the event $A$ with probability $1 - q$. In other words, the probability that $\phi(X^{\ell+1})$ does not decrease by at least a factor of $1 - \frac{1}{4d^2}$ is bounded by the failure probability $1 - \frac{1}{4d^2}$.

Since $\phi(X^{\ell+1}) \leq \phi(X^\ell)$, then under the assumption that $\ell \geq S$ we can upper bound $\sqrt{\phi(X^{\ell+1})}$ with $\sqrt{\phi(X^\ell)}$ in case of failure with probability $q$, and with $\sqrt{(1 - \frac{1}{4d^2})\phi(X^\ell)}$ in case of success

with probability $1 - q$:

$$\mathbb{E}\left[\sqrt{\phi(X^{\ell+1})}|\phi(X^{\ell})\right] \tag{110}$$

$$= \mathbb{E}\left[\sqrt{\phi(X^{\ell+1})}|\phi(X^{\ell}), \bar{A}\right]q + \mathbb{E}\left[\sqrt{\phi(X^{\ell+1})}|\phi(X^{\ell}), A\right](1-q) \tag{111}$$

$$\leq \sqrt{\phi(X^{\ell})}q + \sqrt{\phi(X^{\ell})\left(1 - \frac{1}{4d^2}\right)}(1-q) \tag{112}$$

$$= \sqrt{\phi(X^{\ell})}\left(q + \sqrt{1 - \frac{1}{4d^2}}(1-q)\right) \tag{113}$$

$$= \sqrt{\phi(X^{\ell})}\left(\sqrt{1 - \frac{1}{4d^2}} + \left(1 - \sqrt{1 - \frac{1}{4d^2}}\right)q\right) \qquad \text{monotonic in } q \tag{114}$$

$$\leq \sqrt{\phi(X^{\ell})}\left(\sqrt{1 - \frac{1}{4d^2}} + \left(1 - \sqrt{1 - \frac{1}{4d^2}}\right)\left(1 - \frac{1}{4d^2}\right)\right) \qquad \text{plug } q \leq 1 - \frac{1}{4d^2} \tag{115}$$

$$= \sqrt{\phi(X^{\ell})}\left(1 - \frac{1}{4d^2} + \sqrt{1 - \frac{1}{4d^2}}\frac{1}{4d^2}\right) \qquad \text{rearranging terms} \tag{116}$$

$$\leq \sqrt{\phi(X^{\ell})}\left(1 - \frac{1}{4d^2} + \left(1 - \frac{1}{8d^2}\right)\frac{1}{4d^2}\right) \qquad \sqrt{1-x} \leq 1 - \frac{x}{2} \text{ for } x \geq 0 \tag{117}$$

$$= \sqrt{\phi(X^{\ell})}\left(1 - \frac{1}{32d^4}\right) . \tag{118}$$

Thus, for $\ell \geq S$, we have

$$\mathbb{E}\sqrt{\phi(X^{\ell+1})} = \mathbb{E}_{X^{\ell}}\mathbb{E}\left[\sqrt{\phi(X^{\ell+1})}|\phi(X^{\ell})\right] \leq \mathbb{E}\sqrt{\phi(X^{\ell})}\left(1 - \frac{1}{32d^4}\right) . \tag{119}$$

The summation starts from below $\sqrt{d\phi(X_S)}$, and will decay by rate $1 - \frac{1}{32d^4}$, which is upper bounded by the geometric series:

$$\sqrt{d\phi(X_S)}\sum_{k=0}^{L}\left(1 - \frac{1}{32d^4}\right)^k \leq \sqrt{d\phi(X_0)}32d^4\left(1 - \left(1 - \frac{1}{32d^4}\right)^{L+1}\right) \tag{120}$$

$$\leq 32d^{4.5}\phi(X_0)^{0.5}\left(1 - e^{-\frac{L}{32d^4}}\right) . \tag{121}$$

$$\square$$

*Proof of Lemma D.5.* By Lemma C.2 we have

$$\Pr\left\{\phi(X^{\ell}) \geq \frac{1}{16d}\right\} \leq \exp\left(-\frac{\ell}{\max(2d^2, 32d^3\phi(X^0))}\right)16d\phi(X^0) . \tag{122}$$

Thus, we have

$$\Pr\{S \geq \ell\} \leq \exp\left(-\frac{\ell}{\max(2d^2, 32d^3\phi(X^0))}\right)16d\phi(X^0) . \tag{123}$$

Since $S$ is a non-negative integer valued random variable, we can thus bound $\mathbb{E}[S]$ as:

$$\mathbb{E}[S] = \sum_{\ell=1}^{\infty} P\{S \geq \ell\} \tag{124}$$

$$\leq 16d\phi(X^0) \sum_{\ell=1}^{\infty} \exp\left(\frac{-\ell}{k}\right) \tag{125}$$

$$= 16d\phi(X^0) \frac{1}{\exp(\frac{1}{k}) - 1} \tag{126}$$

$$\leq 16d\phi(X^0)k \tag{127}$$

$$= 16d\phi(X^0) \max(2d^2, 32d^3\phi(X^0)) \tag{128}$$

$$\lesssim 512d^4\phi(X^0)^2 . \tag{129}$$

$\square$

## 1 PROOF OF LEMMA D.4: BOUNDING BN GRAD-NORM WITH ISOMETRY GAP

The proof of the Lemma relies on two main observations that are crystallized in the following lemmas that first establish a bound on Jacobian operator norm based on the inverse of smallest eigenvalue, and then establish a lower bound for the smallest eigenvalue using the isometry gap.

**Lemma D.7.** *Let $X \in \mathbb{R}^{d \times d}$ and let $\{\lambda_i\}_{i=1}^d$ be the eigenvalues of $XX^\top$. Then, we have that:*

$$\|J_{\mathrm{BN}}(X)\|_{op}^2 \leq \frac{1}{\lambda_d}, \tag{130}$$

*where $J_{\mathrm{BN}}$ is the Jacobian of the $\mathrm{BN}(\cdot)$ operator.*

Using the above lemma we have $\log\|J_{\mathrm{BN}}(X)\|_{op} \leq -\log\lambda_d$. The following lemma upper bounds this quantity using isometry gap:

**Lemma D.8.** *The minimum eigenvalue of a Gram matrix that is trace-normalized is lower-bounded by the isometry gap as $-\log\lambda_d \leq d\phi(X) + 1$. Furthermore, if $\phi(X) \leq \frac{1}{16d}$, then $-\log\lambda_d \leq 2\sqrt{d\phi(X)}$.*

Plugging these two values we have the bounds

$$\log\|J_{\mathrm{BN}}(X)\|_{op} \leq d\phi(X) + 1, \tag{131}$$

$$\phi(X) \leq \frac{1}{16d} \implies \log\|J_{\mathrm{BN}}(X)\|_{op} \leq 2\sqrt{d\phi(X)} . \tag{132}$$

Now we can turn our attention to the proof of the Lemmas used in the proof. The proof of relationship between minimum eigenvalue and isometry gap is obtained by merely a few numerical inequalities:

*Proof of Lemma D.8.* Let $\{\lambda_i\}_{i=1}^d$ be the eigenvalues of $X^\top X$. Since the matrix is trace-normalized, we have $\sum_{k=1}^d \lambda_k = d$.

The arithmetic mean of the top $d-1$ values can be written as

$$\frac{1}{d-1} \sum_{k=1}^{d-1} \lambda_k = 1 + \frac{1 - \lambda_d}{d - 1} . \tag{133}$$

Thus, we have that their geometric mean is bounded by the same value. Therefore, we have the following bound:

$$\prod_{k=1}^d \lambda_k \leq \lambda_d \left(1 + \frac{1 - \lambda_d}{d - 1}\right)^{d-1} \tag{134}$$

$$\implies d\log\mathcal{I}(X) \leq \log(\lambda_d) + (d-1)\log\left(1 + \frac{1 - \lambda_d}{d - 1}\right), \tag{135}$$

where in the second inequality we have taken logarithm of both sides. Now, we can apply the numerical inequality $\log(1 + x) \le x$ to conclude:

$$-d\phi(X) \le \log \lambda_d + 1 - \lambda_d . \tag{136}$$

Since $\lambda_d$ is non-negative, this clearly implies the first inequality: $-\log \lambda_d \le d\phi(X) + 1$.

For the second inequality first we use the numerical inequality $\log(x) + 1 - x \le -\frac{(x-1)^2}{2}, \forall x \in [0, 1]$ to conclude

$$\frac{(1 - \lambda_d)^2}{2} \le d\phi(X) \tag{137}$$

$$\implies \lambda_d \ge 1 - \sqrt{2d\phi(X)} \tag{138}$$

$$\implies -\log \lambda_d \le -\log(1 - \sqrt{2d\phi(X)}) . \tag{139}$$

We can now use the inequality $-\log(1 - x) \le \sqrt{2}x$ for $0 \le x \le \frac{1}{2}$ to conclude that

$$-\log \lambda_d \le 2\sqrt{d\phi(X)} \tag{140}$$

when $2\sqrt{d\phi(X)} \le \frac{1}{2}$, which is equivalent to $\phi(X) \le \frac{1}{16d}$. $\qquad\square$

For proving Lemma D.4, we first analyze BN operator on a row, and then invoke this bound and the special structure $J_{\mathrm{BN}}$ to derive the main proof.

**Lemma D.9.** *Let $f : \mathbb{R}^d \to \mathbb{R}^d$ defined as $f(x) = \frac{x}{\|x\|}$ be the elementwise normalization of the $x$. Then:*

$$J_f(x) = \frac{1}{\|x\|} I_{d^2} - \frac{1}{\|x\|^3} x \otimes x, \tag{141}$$

*where $\otimes$ is the outer product.*

*Proof.* To begin, notice that for $x \in \mathbb{R}^d$ we have $\frac{\partial \|x\|}{\partial x} = \frac{x}{\|x\|}$. Denote by $y_i := [f(x)]_i = \frac{x_i}{\|x\|}$. Then the Jacobian entries become:

$$\frac{\partial y_i}{\partial x_i} = \frac{\|x\| - \frac{x_i}{\|x\|} x_i}{\|x\|^2} = \frac{1}{\|x\|} - \frac{1}{\|x\|^3} x_i x_i, \tag{142}$$

$$\frac{\partial y_i}{\partial x_j} = \frac{-\frac{x_j}{\|x\|} x_i}{\|x\|^2} = -\frac{1}{\|x\|^3} x_i x_j . \tag{143}$$

Assembling the equations into matrix form, we obtain:

$$J_f(x) = \frac{1}{\|x\|} I_{d^2} - \frac{1}{\|x\|^3} x \otimes x . \tag{144}$$

$\qquad\square$

**Corollary D.10.** *The $J_f(x)$ has the eigenvalue $\frac{1}{\|x\|}$ with multiplicity $d^2 - 1$ and $0$ with multiplicity 1.*

**Lemma D.11.** *Let $XX^\top = \sum_{i=1}^d \lambda_i u_i u_i^\top$, where $XX^\top = U\Lambda U^\top$ is the eigendecomposition of $XX^\top$. Then, we have that $\min_j \|X_{j\cdot}\|^2 \ge \min_k \lambda_k$.*

*Proof.*

$$\|X_{j\cdot}\|^2 = (XX^\top)_{jj} = \sum_{i=1}^d \lambda_i u_{ji}^2 \ge \min_k \lambda_k \sum_{i=1}^d u_{ji}^2 = \min_k \lambda_k, \tag{145}$$

where in the last equality we have used the fact that $U$ orthogonal. Since this is true for all rows $j$, we get $\min_j \|X_{j\cdot}\|^2 \ge \min_k \lambda_k$. $\qquad\square$

Having established the above properties, we are equipped to prove that the Jacobian of a BN layer is bounded by the inverse of the minimum eigenvalue of its input Gram matrix.

*Proof of Lemma D.7.* To begin, notice that since $\text{BN} : \mathbb{R}^{d^2} \to \mathbb{R}^{d^2}$, implies that $J_{\text{BN}}(X) \in \mathbb{R}^{d^2 \times d^2}$. Denote $X' = \text{BN}(X)$. Since the normalization happens on each row independently of the other rows, the only non-zero derivatives in the Jacobian correspond to changes in output row $i$ with regards to the same input row $i$, i.e.:

$$\frac{\partial X'_{ik}}{\partial X_{jl}} = 0, \quad \forall i \neq j, \forall k, l . \tag{146}$$

This creates a block-diagonal structure in $J_{\text{BN}}(X)$, with $d$ blocks on the main diagonal, where each block has size $d \times d$ and is equal to $J_f(X_{i\cdot})$, where $f$ is as defined in Lemma D.9. Therefore, due to the block-diagonal structure, we know that:

$$\lambda\left[J_{\text{BN}}(X)\right] = \bigcup_{i=1}^{d} \lambda\left[J_f(X_{i\cdot})\right], \tag{147}$$

where $\lambda[\cdot]$ denotes the eigenvalue spectrum. From Corollary D.10, we know that

$$\lambda\left[J_{\text{BN}}(X)\right] = \left\{\frac{1}{\|X_{i\cdot}\|}\right\}_{i=1}^{d} \cup \{0\} \tag{148}$$

with their respective multiplicites. Finally, using Lemma D.11, this implies:

$$\|J_{\text{BN}}(X)\|_2^2 = \left[\max_i \frac{1}{\|X_{i\cdot}\|}\right]^2 = \left[\frac{1}{\min_i \|X_{i\cdot}\|}\right]^2 \leq \frac{1}{\min_k \lambda_k} . \tag{149}$$

$\square$

*Proof of Lemma D.2.* Assuming the the logits are passed through a softmax layer, we analyze the case of Cross Entropy Loss for one sample $i$ in a $C$-classes classification problem. Denoting $z_i = H_{i\cdot}^L$, we have:

$$\mathcal{L}(z_i, y_i) = -\sum_{i=1}^{C} y_i \log p_i, \tag{150}$$

where $p_i = \frac{e^{z_i}}{\sum_{j=1}^{C} e^{z_j}}$ is the probability vector for sample $i$ after passing through the softmax function.

Computing the partial derivatives, we obtain:

$$\frac{\partial p_i}{\partial z_i} = p_i(1 - p_k), \quad i = k \tag{151}$$

$$\frac{\partial p_i}{\partial z_k} = -p_i p_k, \quad i \neq k . \tag{152}$$

Finally, we can compute the gradient of the loss with respect to the logits:

$$\nabla_{z_k}\mathcal{L} = \sum_{i=1}^{C} \left(-y_i \frac{\partial \log(p_i)}{\partial z_k}\right) \tag{153}$$

$$= \sum_{i=1}^{C} \left(-y_i \frac{1}{p_i} \frac{\partial p_i}{\partial z_k}\right) \tag{154}$$

$$= \sum_{i \neq k} (y_i p_k) + (-y_k(1 - p_k)) \tag{155}$$

$$= p_k \sum_{i \neq k} y_i - y_k(1 - p_k) \tag{156}$$

$$\implies \|\nabla_{z_k}\mathcal{L}\| \leq \left\|p_k \sum_{i \neq k} y_k\right\| + \|y_k(1 - p_k)\| \leq 2 . \tag{157}$$

Since the gradient of the loss with regard to each sample is bounded, we can conclude that the operator norm of the Jacobian of the loss with regards to the logits matrix $H^L$ is also bounded.

In a similar analysis, we now shift our attention towards the Mean Squared Error (MSE) loss:

$$\mathcal{L}(z_i, y_i) = \frac{1}{C} \sum_{i=1}^{C} (y_i - p_i)^2 .$$

We want to compute the gradient of the loss with respect to each logit $z_k$:

$$\|\nabla_{z_k} \mathcal{L}\| = \frac{1}{C} \sum_{i=1}^{C} 2(y_i - p_i) \frac{\partial(-p_i)}{\partial z_k} \tag{158}$$

$$\implies \|\nabla_{z_k} \mathcal{L}\| \leq \frac{2}{C} \sum_{i=1}^{C} \left| (y_i - p_i) \frac{\partial p_i}{\partial z_k} \right| \leq \frac{2}{C} \sum_{i=1}^{C} |y_i - p_i| \cdot \left| \frac{\partial p_i}{\partial z_k} \right| \leq 2 . \tag{159}$$

By substituting these derivatives into the gradient equation, we can derive the gradient for each logit with respect to the MSE loss.

$\square$

**Lemma D.12.** *Let $X^\ell \in \mathbb{R}^{d \times d}$ be the hidden representations of layer $\ell > 0$ as defined in equation 3. Then, we have that:*

$$\log \left\| \frac{\partial X^{\ell+1}}{\partial X^\ell} \right\| \leq \log \left\| J_{\text{BN}}(H^\ell) \right\| . \tag{160}$$

*Proof of Lemma D.12.* By definition, we have that:

$$H^\ell = W^\ell X^\ell, \tag{161}$$

$$X^{\ell+1} = \text{BN}(H^\ell) . \tag{162}$$

Therefore, applying the chain rule, we get:

$$\frac{\partial X^{\ell+1}}{\partial X^\ell} = \frac{\partial X^{\ell+1}}{\partial H^\ell} \frac{\partial H^\ell}{\partial X^\ell} = J_{\text{BN}}(H^\ell) W^\ell . \tag{163}$$

Taking the logarithm of the norm of this quantity, we reach the conclusion:

$$\log \left\| \frac{\partial X^{\ell+1}}{\partial X^\ell} \right\| \leq \log \left\| J_{\text{BN}}(H^\ell) \right\| + \log \left\| W^\ell \right\| = \log \left\| J_{\text{BN}}(H^\ell) \right\|, \tag{164}$$

where we have used the fact that the spectrum of the orthogonal matrix $W^\ell$ contains only the singular value 1 with multiplicity $d$. $\square$

# E LINEAR INDEPENDENCE IN COMMON DATASETS

In this section, we empirically verify the assumption that popular datasets do not suffer from rank collapse in most practical settings.

We provide empirical evidence for CIFAR10, MNIST, FashionMNIST and CIFAR100. We test this assumption by randomly sampling 100 input batches of sizes $n = 16, 32, 64, 128, 256, 512$ from each of these datasets and then measuring the rank of the Gram matrix of these randomly sampled batches using the `matrix_rank()` function provided in PyTorch. We stop at size 512 since we approach the dimensionality of some datasets, i.e. FashionMNIST, MNIST. We show in Table E1 the average rank with the standard deviation for each $n$, over 100 randomly sampled batches.

Table E1: Average rank of Gram matrix of input batches of size $n$ from different datasets. Mean and standard deviation are computed over 100 randomly selected input batches, where the samples are chosen without replacement.

| Dataset | $n = 16$ | $n = 32$ | $n = 64$ | $n = 128$ | $n = 256$ | $n = 512$ |
|---|---|---|---|---|---|---|
| CIFAR10 | $16.0 \pm 0.0$ | $32.0 \pm 0.0$ | $64.0 \pm 0.0$ | $127.99 \pm 0.09$ | $221.06 \pm 2.89$ | $203.70 \pm 3.58$ |
| MNIST | $16.0 \pm 0.0$ | $32.0 \pm 0.0$ | $64.0 \pm 0.0$ | $128.00 \pm 0.00$ | $250.48 \pm 1.53$ | $318.04 \pm 2.82$ |
| FashionMNIST | $16.0 \pm 0.0$ | $32.0 \pm 0.0$ | $64.0 \pm 0.0$ | $128.00 \pm 0.00$ | $238.19 \pm 3.27$ | $275.37 \pm 4.20$ |
| CIFAR100 | $16.0 \pm 0.0$ | $32.0 \pm 0.0$ | $64.0 \pm 0.0$ | $127.92 \pm 0.27$ | $218.11 \pm 3.69$ | $201.51 \pm 3.95$ |

We would like to remark that these datasets are fairly simple in terms of dimensionality and semantics, which can lead to correlated samples. Furthermore, the rank degeneracy can be alleviated even in the larger batch sizes through various data augmentations techniques. Note that these datasets have a high degree of correlation between samples. Most notably, the average cosine similarity between samples in a 512 size batch is $0.81, 0.40, 0.58, 0.81$ for CIFAR10, MNIST, FashionMNIST and CIFAR100 respectively.

# F    ACTIVATION SHAPING

In this section, we explain the full procedure for shaping the activation, as well as expand on the heuristic we use to choose the pre-activation gain. Under the functional structure of the MLP in equation 11, let $\alpha_\ell$ be the pre-activation gain.

More formally, since the gradient norm has an exponential growth in depth, as shown in Figure H5, we can compute the linear growth rate of log-norm of gradients in depth. We define the rate of explosion for a model of depth $L$ and gain $\alpha$ at layer $\ell$ as the slope of the log norm of the gradients:

$$R(\ell, \alpha_\ell) = \frac{\log \|\nabla_{W_\ell} \mathcal{L}\| - \log \|\nabla_{W_{\ell-10}} \mathcal{L}\|}{10} \ . \tag{165}$$

Since the rate function is not perfectly linear and has noisy peaks, we measure the slope with a 10 layer gap in order to capture the true behaviour instead of the noise.

Our goal is to choose $\alpha$ such that the sum of the rates across the layers in depth is bounded by a constant that does not depend on the depth of the model, i.e. $R(\ell, \alpha_\ell) \leq \beta$, where $\beta$ is independent of $L$. One choice to achieve this is to pick a gain such that the sum of the rates behaves like a decaying harmonic sum in depth.

To this end, we measure the rate of explosion at multiple layers in a 1000 layer deep model, for various gains $\alpha$ which are constant across the layers in Figure F1 and notice that it behaves as $R(\ell) = c_1 \alpha_2^c$. In order to have the sum of rates across layers behave like a bounded harmonic series in depth, we must choose the gain such that it decays roughly as $\alpha^{c_2} = \ell^{-k}$ where $k > 1$ results in convergence. Therefore, we can obtain a heuristic for picking a gain such that the gradients remain bounded in depth as $\alpha_\ell = \ell^{-k/c_2}$, where we refer to $k/c_2$ as the gain exponent.

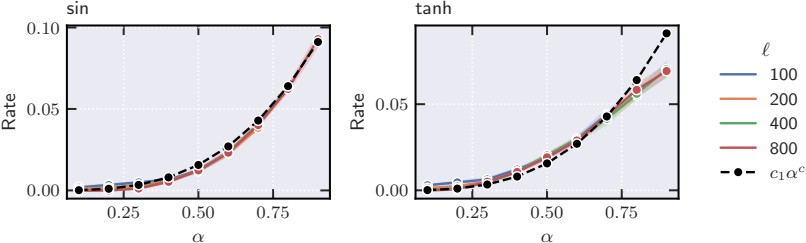

Figure F1: Explosion rate of the log norm of the gradients at initialization for an MLP model with orthogonal weights and batch normalization, for sin and tanh nonlinearities measured for a 1000 layer deep model at layers $\ell$ as a function of gain $\alpha$. The black trace shows the fitted function $c_1 \alpha_2^c$. Traces are averaged over 10 independent runs, with the shades showing the 95% confidence interval.

This reduces the problem to picking the exponent such that the sum stays bounded. We show how the behaviour of the explosion rate at the early layers, for various models, is impacted by the exponent in Figure F2. Note that for several exponent values, we able to reduce the exponential explosion rate and obtain trainable models, which we show in Appendix H.

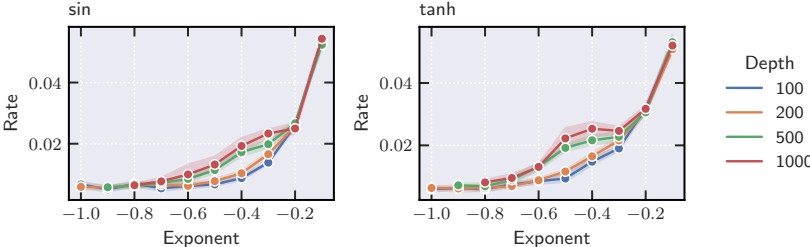

Figure F2: Explosion rate of the log norm of the gradients at initialization for an MLP model with orthogonal weights and batch normalization, for sin and tanh nonlinearities at depths $100, 200, 500, 1000$ as a function of the gain exponent. Traces are averaged over $10$ independent runs, where the shade shows the $95\%$ confidence interval. Rate is measured at $\ell = 10$ to avoid the any transient effects of the function

## G IMPLICIT ORTHOGONALITY DURING TRAINING

In this section, we provide empirical evidence that our architecture during training maintains orthogonality across depths, while maintaining bounded gradients. Figure G1 shows the evolution of the isometry gap of the weight matrices $W_\ell$ during training, for models at different depths and different nonlinearities. In order to show that these weights are updated gradient descent, we also show the evolution of the norm of the loss gradients with regards to matrices $W_\ell$ in Figure G2.

These experiments are performed on an MLP with orthogonal weight matrices and batch normalization, with sin and tanh activations. The width is set to 100, batch size 100 and learning rate 0.001. The gain exponent is set to a fixed value for all experiments. The measurements are performed on a single batch of size 100 from CIFAR10, after each epoch of training on the same dataset.

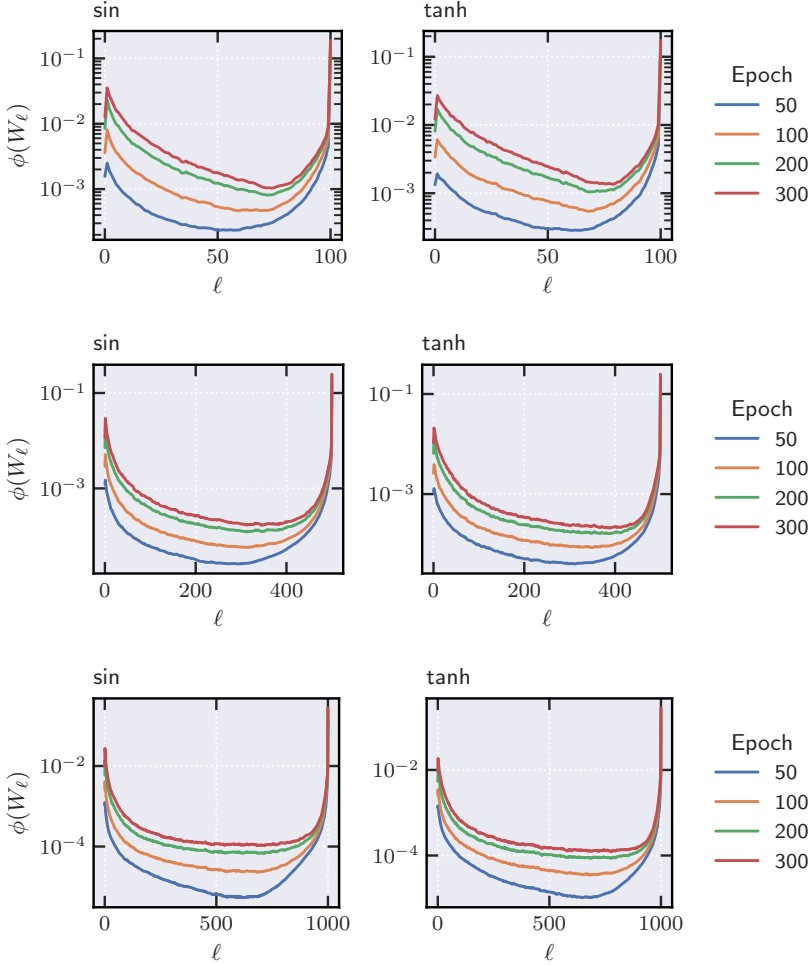

Figure G1: Contrasting the isometry gap of weight matrices during training for MLPs of depth 100 (top), 500 (middle), 1000 (bottom). The middle layers become increasingly more orthogonal with depth, while maintaining a small isometry gap. During training, the isometry gap also remains low, suggesting the matrices remain close to being orthogonal.

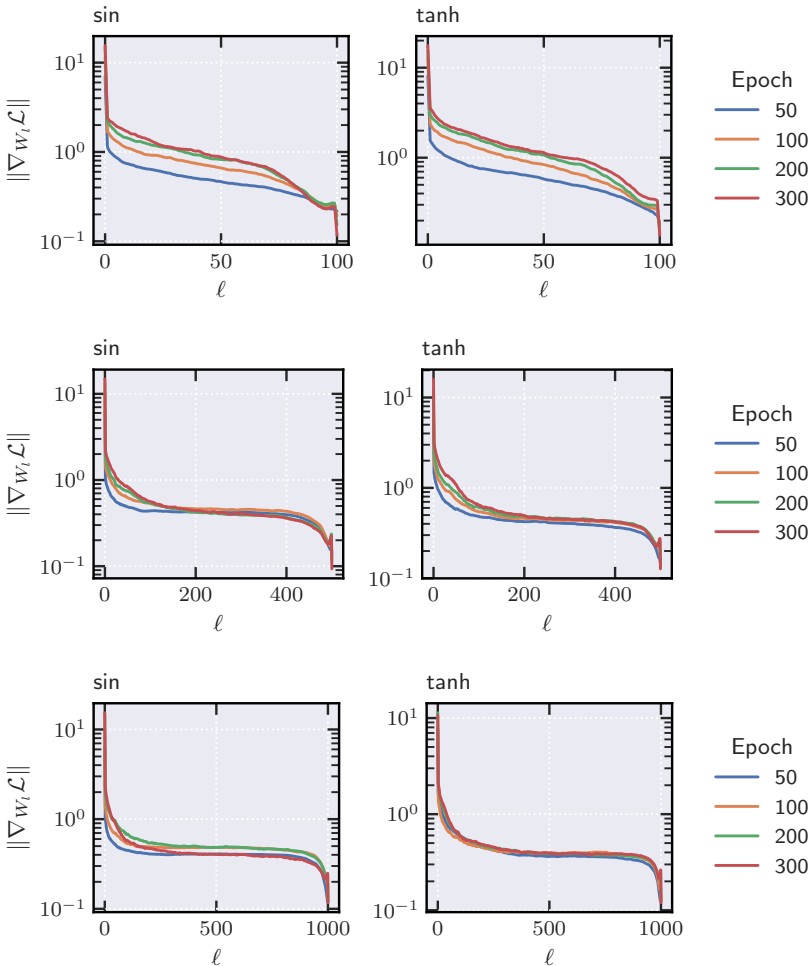

Figure G2: Contrasting the Frobenius norm of the gradients of the loss with respect to the weights during training for MLPs of depth 100 (top), 500 (middle), 1000 (bottom). The gradients do not vanish during training and across different depths for all layers, suggesting that the orthogonality evidenced in Figure G1 is not due to the weights not being updated during SGD.

# H    OTHER EXPERIMENTS

In this section we provide the train and test accuracies of deep MLPs on 4 popular image datasets, namely MNIST, FashionMNIST, CIFAR10, CIFAR100. Hyperparameters and measurements procedure are described in Section 4.

## 1    SUPPLEMENTARY TRAIN AND TEST RESULTS ON MNIST, FASHIONMNIST, CIFAR10, CIFAR100

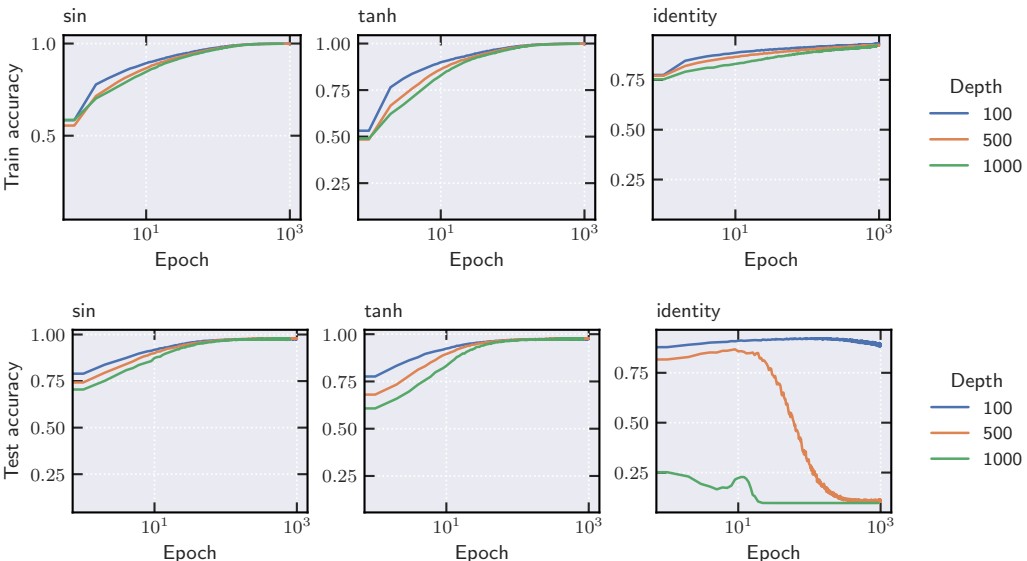

Figure H1: Contrasting the train and test accuracy of MLPs with gained sin, tanh and identity activations on MNIST. The identity activation performs much worse than the nonlinearities, indicating the fact that the sin and tanh networks are not operating in the linear regime. The network is trained with vanilla SGD and the hyperparameters are width 100, batch size 100, learning rate 0.001.

## 2    SUPPLEMENTAL FIGURES

We present empirical results in Figure 2 showing that degenerate input batches are a hard constraint for orthogonalization without gradient explosion. For MLPs with different depths, we show that by repeating samples in a batch of size 10 we get an exponential gradient explosion, which is unavoidable theoretically.

Furthermore, we show how non-linearities affect the gradient explosion rate in Figure H5. Using standard batch normalization and fully connected layers from PyTorch we show that non-linearities maintain a large isometry gap. This is a critical issue for our theoretical framework, since we take advantage of the fact that the identity activation achieves perfect orthogonality in order to prove that the gradients remain bounded in depth.

## 3    INFLUENCE OF MEAN REDUCTION ON THE GRADIENT BOUND

In this section, we compare whether adding mean reduction and the additional factor of $\frac{1}{n}$ in the denominator of the batch normalization module influences our gradient bound. As expected, we show in Figure H6 that in both cases, for the identity activation, the result remains similar, with the gradients remaining bounded in depth.

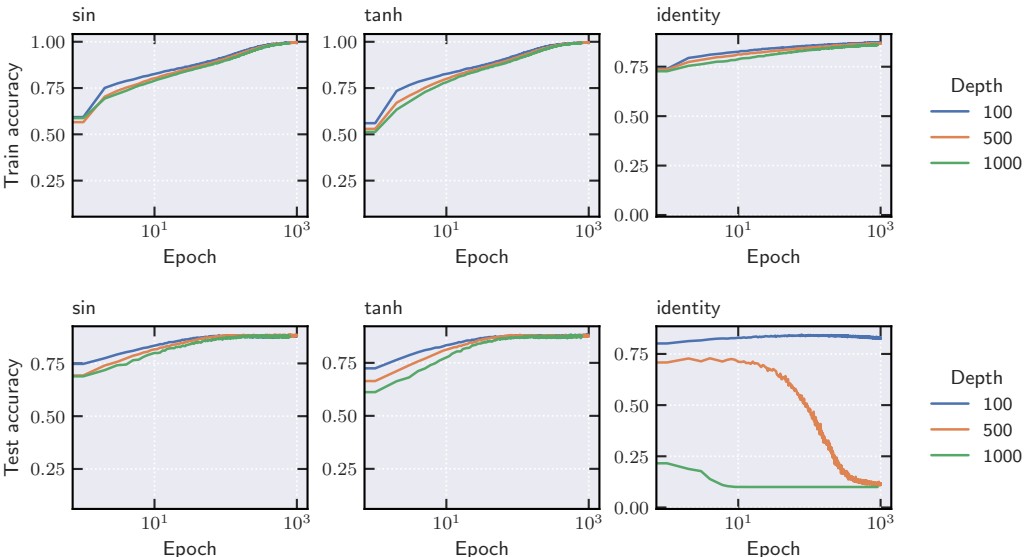

Figure H2: Contrasting the train and test accuracy of MLPs with gained sin, tanh and identity activations on FashionMNIST. The identity activation performs much worse than the nonlinearities, indicating the fact that the sin and tanh networks are not operating in the linear regime. The networks are trained with vanilla SGD and the hyperparameters are width 100, batch size 100, learning rate 0.001.

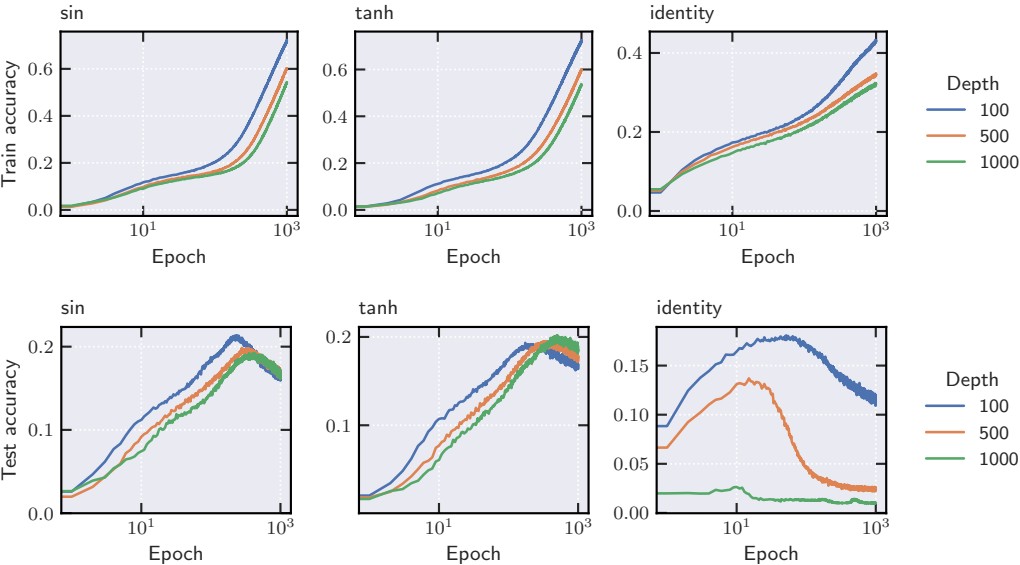

Figure H3: Contrasting the train and test accuracy of MLPs with gained sin, tanh and identity activations on CIFAR100. The identity activation performs much worse than the nonlinearities, indicating the fact that the sin and tanh networks are not operating in the linear regime. The networks are trained with vanilla SGD and the hyperparameters are width 100, batch size 100, learning rate 0.001.

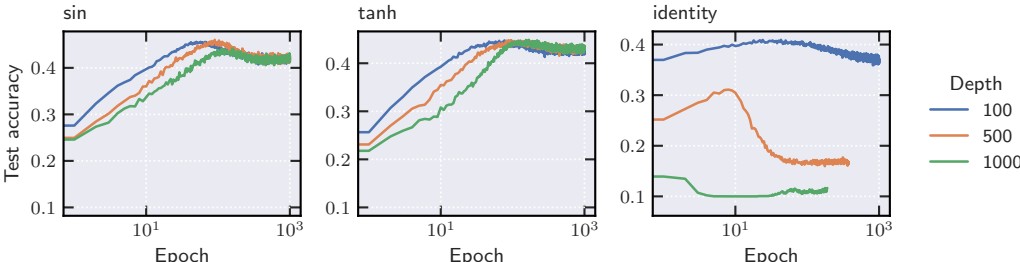

Figure H4: Contrasting the train test accuracy of MLPs with gained sin, tanh and identity activations on CIFAR10. The identity activation performs much worse than the nonlinearities, indicating the fact that the sin and tanh networks are not operating in the linear regime. The networks are trained with vanilla SGD and the hyperparameters are width 100, batch size 100, learning rate 0.001.

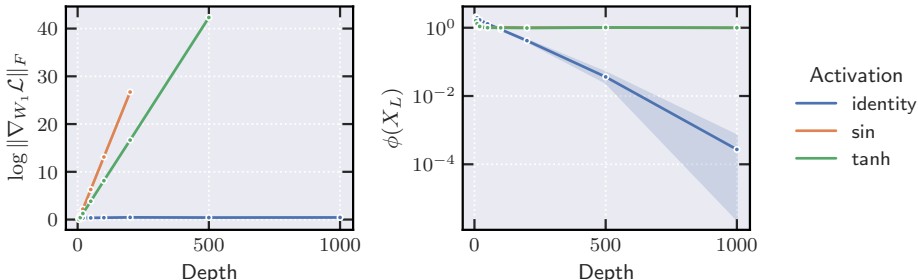

Figure H5: Left: average log-norm of gradients (log-scale y-axis) at the first layer for networks with different depths, evaluted on CIFAR10. Right: Isometry gap (log-scale y-axis) at the last layer for networks with different depths, evaluted on CIFAR10. The MLP is initialized with orthogonal weights and batch normalization, with standard modules, with sin, tanh, identity non-linearities. After stabilizing the isometry gap, the non-linearities have an exponential gradient explosion.

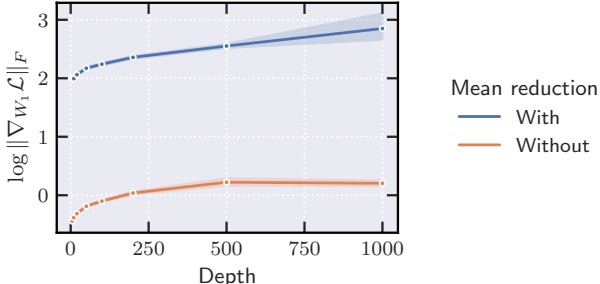

Figure H6: Comparing the gradient explosion rate in networks with standard batch normalization (blue) and networks with the simplified batch normalization operator from our theoretical framework (orange). Notice that the 2 traces are similar in terms of gradient explosion. Traces are averaged over 10 runs with the shaded regions showing the 95% confidence interval. Samples are from CIFAR10.

## I  HIGH PROBABILITY LOG-GRADIENT BOUND

**Theorem I.1.** *In the same conditions as Theorem D.1, it there is constant $C$ such that:*

$$\Pr\left(\log\|\nabla_{W_\ell}\mathcal{L}\|_{op} \geq C\log(1/\delta)d^6(\phi(X^0)^3 + 1)\right) \leq 2\delta$$

*for any $\delta \in (0, 1)$.*

For example for $\delta = 1/2$, we we have $\|\nabla_{W_\ell}\mathcal{L}\|_{op} \lesssim d^6(\phi(X^0)^3 + 1)$ with at most $1/2$ probability.

*Proof.* We follow the same proof as Theorem D.1 by decomposing the log gradient to sum of log gradients of each layer. Note that Lemma D.4 holds deterministically, as it is valid by construction. Thus, what remains is to derive a probabilistic version of Lemma D.3. We start from equation 100 which holds deterministically, to arrive at a probabilistic bound:

$$\sum_{k=1}^{L} \log \left\| J_{\mathrm{BN}}(H^k) \right\| \leq S(d\phi(X^0)) + 2 \sum_{\ell=S+1}^{L} \sqrt{d\phi(X^\ell)}, \qquad S := \min\left\{ \ell : \phi(X^\ell) \leq \frac{1}{16d^2} \right\}$$
(166)

We derive a probabilistic bound for each term.

**First term.** Based on Lemma D.5 we have $E[S] \lesssim d^4 \phi(X^0)^2$. Thus, there is $C_0$ such that

$$\exists C_0 : E[S] \leq \frac{C_0}{2} d^4 \phi(X^0)^2.$$

Thus, by Markov inequality we have

$$P(S \geq B) \leq \frac{\mathbb{E}[S]}{B} \leq \frac{1}{2}, \qquad B := C_0 d^4 \phi(X^0)^2.$$

First, we discretize the layers into blocks of size $B$, where $B$ is defined above. With a slight abuse of notation, define $B_i$ as the end of the $i$th block of size $B$, and let $E_i$ be the event $E_i = \{\phi(X_{B_i}) > \frac{1}{16d^2}\}$, which is the event of failure for $\phi$ to drop below the threshold of $\frac{1}{16d^2}$ after the last layer from block $i$.

By the inequality established above, we know that in each block, $\phi$ can either fall below the threshold, or stay above, with probability at most $1/2$. Moreover, knowing that $\phi$ is non-increasing, we know that $P(E_i|E_{<i}) \leq 1/2$.

Thus, by the non-increasing property of $\phi$, the probability of failure after $k$ blocks of size $B$ is at most the probability that $\phi$ did not fall below the threshold in any of the $k$ successive blocks:

$$P(E_k) \leq P(E_1 \wedge \cdots \wedge E_{k-1}) \tag{167}$$
$$= P(E_1)P(E_2|E_1)\ldots P(E_k|E_{k-1}\ldots E_1) \tag{168}$$
$$\leq \left(\frac{1}{2}\right)^k \tag{169}$$

Thus, we obtain the probability:

$$P(S \geq kB) = P(S \geq kC_0 d^4 \phi(X_0)^2) \leq 2^{-k} \tag{170}$$

Thus, connecting back to gradients, we obtain:

$$\Pr\left( \sum_{\ell=0}^{S} \|J_{\mathrm{BN}}(H^\ell)\|_{op} \geq kC_0 d^5 \phi(X^0)^3 \right) \leq 2^{-k}. \tag{171}$$

**Second term** Starting from equation 109, we have

$$\ell \geq S \implies \Pr\left\{ \phi(X^{\ell+1}) \geq (1 - 1/4d^2)\phi(X^\ell) \right\} \leq 1 - \frac{1}{4d^2}$$

Let $E_\ell$ denote the event that $\phi(X^{\ell+1})$ does not decrease by $1 - 1/4d^2$ compared to its previous layer: $E_\ell = \mathbf{1}\left\{ \phi(X^{\ell+1}) \geq (1-1/4d^2)\phi(X^\ell) \right\}$. Due to the non-increasing property of $\phi$, we know that $\Pr(E_\ell) = \Pr(E_\ell|E_{\ell-1})$. Repeating this for $s$ steps, we obtain:

$$\ell \geq S \implies \Pr\left\{ \phi(X^{\ell+s}) \geq (1 - 1/4d^2)\phi(X^\ell) \right\} = \prod_{i=\ell}^{\ell+s-1} \Pr\{E_{i+1}|\bar{E}_i\} \leq (1 - \frac{1}{4d^2})^s \quad (172)$$

Inspired by this, consider the sequence $\ell_0, \ell_1, \ldots$ defined as

$$\ell_0 = S, \qquad \ell_k = \ell_{k-1} + 4ckd^2$$

Thus, we get:

$$\Pr\left\{\phi(X^{\ell_{k+1}}) \geq (1 - \frac{1}{4d^2})\phi(X^{\ell_k})\right\} \leq (1 - \frac{1}{4d^2})^{4ckd^2} \leq e^{-ck}$$

Thus, we have the union bound

$$\Pr\left\{\bigvee_{k=0}^{\infty} \phi(X^{\ell_{k+1}}) \geq (1 - \frac{1}{4d^2})\phi(X^{\ell_k})\right\} \leq \sum_{k=1}^{\infty} e^{-ck} = \frac{e^{-c}}{1 - e^{-c}}$$

$$\implies \Pr\left\{\underbrace{\bigwedge_{k=0}^{\infty} \phi(X^{\ell_{k+1}}) \leq (1 - \frac{1}{4d^2})\phi(X^{\ell_k})}_{Q:=}\right\} \geq 1 - \frac{e^{-c}}{1 - e^{-c}} = \frac{1 - 2e^{-c}}{1 - e^{-c}}$$

Note that in the event that $Q$ holds, we can the isometry gaps in each $[\ell_{k-1}, \ell_k]$ interval as:

$$Q \implies \phi(X_\ell) \leq \frac{1}{16d}(1 - \frac{1}{4d^2})^{k-1} \text{ for all } \ell \in [\ell_{k-1}, \ell_k)$$

We can upper bound $\sum_{\ell=S+1}^{\infty} \sqrt{d\phi(X^\ell)}$ by using the numbers of items and upper bound on each block. Thus, assuming for all $k$ we have $\phi(X^{\ell_{k+1}}) \leq (1 - \frac{1}{4d^2})\phi(X^{\ell_k})$, we can derive

$$Q \implies \sum_{\ell=S+1}^{\infty} \sqrt{d\phi(X_\ell)} \leq \sum_{k=1}^{\infty}(\ell_k - \ell_{k-1})\sqrt{d\frac{1}{16d}(1 - \frac{1}{4d^2})^k}$$

$$= cd^2 \sum_{k=1}^{\infty} k(1 - \frac{1}{4d^2})^{k/2}$$

$$\leq cd^2 \sum_{k=1}^{\infty} k(1 - \frac{1}{8d^2})^k \qquad \text{using } (1 + x)^{1/2} \leq 1 + x/2$$

$$= cd^2 \frac{1 - 8/d^2}{(1/8d^2)^2} \qquad \text{using } \sum_{k=1}^{\infty} k\alpha^k = \alpha/(1 - \alpha)^2.$$

$$\leq 64cd^6$$

Thus we have:

$$\Pr\left(\sum_{\ell=S+1}^{\infty} \sqrt{d\phi(X_\ell)} \leq 64cd^6\right) \geq \Pr\left\{Q\right\} \geq \frac{1 - 2e^{-c}}{1 - e^{-c}}$$

which yields

$$\Pr\left(\sum_{\ell=S+1}^{\infty} \log\|J_{BN}(H^\ell)\|_{op} \geq 64cd^6\right) \leq \Pr(\overline{Q}) = \frac{e^{-c}}{1 - e^{-c}}$$

Combining first and second term bounds for the Jacobian log-norms we have

$$\Pr\left(\sum_{l=\ell}^{S} \log\|J_{\text{BN}}(H^l)\|_{op} > kC_0 d^5 \phi(X^0)^3\right) \leq 2^{-k} \tag{173}$$

$$P\left(\sum_{\ell=S+1}^{\infty} \log\|J_{\text{BN}}(H^l)\|_{op} \geq 64cd^6\right) \leq \frac{e^{-c}}{1 - e^{-c}} \tag{174}$$

And thus we have

$$\Pr\left(\sum_{l=\ell}^{L} \log\|J_{BN}(H^l)\|_{op} \geq kC_0 d^5 \phi(X^0)^3 + 64cd^6\right) \leq 2^{-k} + \frac{e^{-c}}{1 - e^{-c}}$$

Thus, we can find $C$ such that

$$\Pr\left(\log\|\nabla_{W_\ell}\mathcal{L}\|_{op} \geq kCd^6(\phi(X^0)^3 + 1)\right) \leq 2^{-k+1}$$

We can finish the proof by defining $\delta = 2^{-k}$ and change of variables $k = \log(1/\delta)$. $\qquad\square$

## J   GRADIENTS IN RESIDUAL NETWORKS

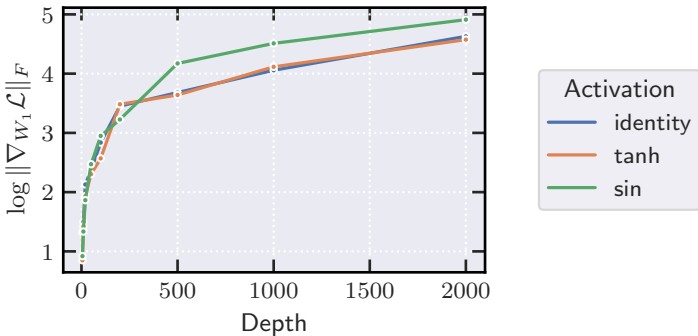

Figure J1: Logarithmic plot for the gradient norm of the first layer for residual networks with batch normalization initialized with Gaussian weights at different depths, evaluated on CIFAR10. The traces show that the gradients do not remain bounded in depth for different activations.

