## A    CONDITIONAL ORTHOGONALIZATION

**Isometry after rotation.**    Our analysis is based on  (Joudaki et al., 2023b, Corollary 3), which we restate in the following Lemma:

**Lemma A.1.** *For all non-degenerate matrices $X \in \mathbb{R}^{d \times d}$, we have:*

$$\mathcal{I}(\text{BN}(X)) \geq \left( 1 + \frac{variance\{\|X_{j\cdot}\|\}_{j=1}^d}{(mean\{\|X_{j\cdot}\|\}_{j=1}^d)^2} \right) \mathcal{I}(X) \,. \tag{12}$$

Lemma A.1 proves isometry bias of BN.  The next lemma proves that isometry does not change under rotation

**Lemma A.2** (Isometry after rotation)**.** *Let $X \in \mathbb{R}^{d \times d}$ and $W \sim \mathbb{O}_d$ be a random orthogonal matrix and $X' = WX$. Then,*

$$\mathcal{I}(\text{BN}(X')) \geq \left( 1 + \frac{variance\{\|X'_{j\cdot}\|\}_{j=1}^d}{(mean\{\|X'_{j\cdot}\|\}_{j=1}^d)^2} \right) \mathcal{I}(X) \,. \tag{13}$$

*Proof.*  Using properties of the determinant, we have

$$\det(X'X'^\top) = \det(W)^2 \det(XX^\top) = \det(XX^\top), \tag{14}$$

where the last equation holds since $W$ is an orthogonal matrix. Furthermore,

$$\text{Tr}(X'X'^\top) = \text{Tr}(WXX^\top W^\top) \tag{15}$$

$$= \text{Tr}(XX^\top \underbrace{W^\top W}_{=I}) \tag{16}$$

$$= \text{Tr}(XX^\top) \,. \tag{17}$$

Combining the last two equations with Lemma A.1 concludes the proof.    □

**Increasing isometry with rotations and** BN**.**    The last lemma proves the isometry bias does not decrease with rotation and BN. However, this does not prove a strict decrease in isometry with BN and rotation. The next lemma proves there exists an orthogonal matrix for which the isometry is strictly increasing.

**Corollary A.3** (Increasing isometry)**.** *Let $X \in \mathbb{R}^{d \times d}$ and denote its singular value decomposition $X = U diag\left(\{\sigma_i\}_{i=1}^d\right) V^\top$, where $U$ and $V$ are orthogonal matrices. Then, we have:*

$$\mathcal{I}(\text{BN}(U^\top H)) = 1 \,. \tag{18}$$

*Proof.*  Let $S = \text{diag}\left(\{\sigma_i\}_{i=1}^d\right)$ be the diagonal matrix containing the singular values of $X$. Then, we have:

$$\text{BN}(U^\top X) = \text{BN}(U^\top U S V^\top) \tag{19}$$

$$= \text{BN}(SV^\top) \tag{20}$$

$$= \text{diag}(SV^\top V S)^{-\frac{1}{2}} SV^\top \tag{21}$$