# OpenReview forum: "Towards Training Without Depth Limits: Batch Normalization Without Gradient Explosion"
_ICLR.cc/2024/Conference — ICLR 2024 poster_

### Official Review · Reviewer_uof1 · 2023-10-16

**Soundness:** 3 good
**Presentation:** 3 good
**Contribution:** 2 fair
**Rating:** 6
**Confidence:** 3

**Summary:**

The paper is a theoretical study on batch normalization. A simplified setting is considered where BN does not use mean subtraction and where there are no other non-lineary functions. The paper presents two theorems that hold when the weight matrices are random orthogonal matrices. Theorem 1 states that the representations become increasingly orthogonal with the depth of the network. Theorem 5 states that the gradients do not explode with depth. All results hold in expectation over the random weights. The authors also provide a method for modifying non-linear activation functions by changing the gain.

**Strengths:**

The paper presents two theorems which are relevant and non-asymptotic.

Normalization is used everywhere in ML, so studying it has a high impact.

**Weaknesses:**

The main question of the paper is “Is there any network with batch normalization without gradient explosion and rank collapse issues?”. It is not clear why this question is interesting. The authors show that orthogonal random weights avoid these issues, but such initialization schemes are not used in practice despite being well known. Thus, in practice, BN works for reasons that are unrelated to the specific weight initialization scheme that is proposed. So the paper is not really “explaining” why BN works.

There are still some simplifying assumptions in the theoretical parts – there are no nonlinearities and the mean subtraction of BN is omitted.

It is not clear how relevant the empirical results are, though the authors might be able to clarify this for me.

**Questions:**

Why is the main question (“Is there any network with batch normalization without gradient explosion and rank collapse issues?”) interesting to practioners? Given that BN works well without orthogonal initialization.

How is the activation shaping motivated by theory?

What is the practical utility of the activation shaping you propose?

Does your result hold for networks with mean reduction?

---

> ### Author Response · Authors · 2023-11-17
>
> We appreciate the reviewer's insightful comments on our theoretical work. Moreover, we appreciate the reviewer's great questions and for addressing the impact of our work in practice.
>
> > The main question of the paper is “Is there any network with batch normalization without gradient explosion and rank collapse issues?”. It is not clear why this question is interesting.
>
> Let us elaborate why this question is important and interesting:
>
> *Why is it important?* BN and depth can be beneficial in many learning tasks, as they improve signal propagation and expand the functional class and expressivity of the model. However, if one opts to use batch normalization, they cannot train very deep networks due to gradient explosion. Our study shows that by simply changing the weight distribution from Gaussian to the orthogonal ensemble, training very deep networks becomes feasible. Furthermore, we extend this result to non-linear activations by a specific activation shaping mechanism.
>
> *Why is it interesting?* [1] proves that gradients in MLPs with BN and Gaussian weights explode regardless of the type of activation (including linear). This seems to suggest that gradient explosion with BN is *"unavoidable".* Our study proves that that is not the case.
>
> > The authors show that orthogonal random weights avoid these issues, but such initialization schemes are not used in practice despite being well known.
>
> Initializing weight matrices with elements from the orthogonal ensembles has been used in practice, in some cases to great practical benefits as discussed in the "Initialization with orthogonal matrices" paragraph in our paper.
>
> > Thus, in practice, BN works for reasons that are unrelated to the specific weight initialization scheme that is proposed. So the paper is not really “explaining” why BN works.
>
>  Our paper does not intend to prove benefits of BN, but instead to avoid its issue of gradient explosion. According to [1], the standard initialization scheme of Gaussian weights can not avoid gradient explosion. Under specific hyperparameter choices (batch size of BN) and weight initialization, we prove the explosion is avoidable.
>
> > There are still some simplifying assumptions in the theoretical parts – there are no nonlinearities and the mean subtraction of BN is omitted.
>
> While our theoretical results are limited to linear activations and a simplified BN operator, our theory inspires an activation shaping mechanism that allows us to avoid gradient explosion in networks with activations (Section 5). We theoretically study networks with linear activations due to their analytical simplicity and due to the fact that they also suffer from gradient explosion [1].
>
> > It is not clear how relevant the empirical results are, though the authors might be able to clarify this for me.
>
> Our empirical results elaborated in Sections 4 and 5 show that these techniques enable us to train up to $1000$ layer deep networks with batch normalization. In contrast, using Gaussian weights, training networks with more than $200$ layers is not possible due to gradient explosion.
>
>
> > How is the activation shaping motivated by theory?
>
> The high-level intuition that we draw from our theory is that we can ensure that that the log operator norm of the Jacobian of the activation layers, is upper-bounded by a value that forms a Harmonic series wrt the layer depth. The main connection between our theory and our activation shaping is that if we assume that Thm. 1 holds for an MLP with shaped activation, we can deduce the rest of our results. For an elaborate explanation of the connection between our theory and the activation shaping, we refer the reviewer to appendix Section E.
>
> > What is the practical utility of the activation shaping you propose?
>
> Our activation shaping scheme enables training very deep networks with BN and non-linear activations. Without orthogonal weights or the shaping parameter $\alpha,$ training networks with BN and activation layers is not feasible for very deep (>200 layers) networks, due to gradient explosion. Since deeper networks may have desirable properties (see Supplementary Figure G), this scheme opens up new avenues for neural architecture design.
>
> > Does your result hold for networks with mean reduction?
>
> All the empirical results in Sections 4, 5, and 6, use the standard BN module from PyTorch, which does have mean reduction. While the theory was developed for the BN operator without mean reduction, these empirical results show that the consequences of our theory extend beyond this simplified BN setting. Exploring BN with centering could be a good topic for follow-up studies.
>
> [1] Yang, Greg, et al. "A mean field theory of batch normalization." arXiv preprint arXiv:1902.08129 (2019).

---

> > ### Comment · Reviewer_uof1 · 2023-11-22
> > **reply**
> >
> > I thank the authors for their reply. I will retain my score. Some comments are given below:
> >
> >
> > "However, if one opts to use batch normalization, they cannot train very deep networks due to gradient explosion." -- this depends on the definition of deep, but e.g. Resnets and densenets can be trained relatively deep. Typically, model depth and width are scaled jointly, and I'm not sure the issues with batch normalization not working for extremely deep networks is a practical concern.
> >
> > "Initializing weight matrices with elements from the orthogonal ensembles has been used in practice" -- sorry, I should have been more clear. They are typically not used in practice.

---

> ### Author Response · Authors · 2023-11-23
> **Comments**
>
> We thank the reviewer very much for their comments. We have added Appendix I, showing that for residual networks the gradients are not bounded at arbitrary depth for different activations, including identity.

---

### Official Review · Reviewer_GLi5 · 2023-10-30

**Soundness:** 3 good
**Presentation:** 3 good
**Contribution:** 2 fair
**Rating:** 6
**Confidence:** 4

**Summary:**

The paper addresses limits of gradient descent loss minization of (very) deep networks using batch normalization (BN). In particular, it mends explosion of gradients during backdrop training in such networks. After a neat introduction a novel theoretical results are laid out in an accessible manner and main idea of proofs is presented (linked to full proofs in Appendix). Using recent results on random matrix theory, namely that BN does not decrease isometry gap (a measure of deviance of sample covariance $XX^T$ from orthogonality), paper proofs its main Theorem 1. It claims that expected isometry gap has an upper bound whose rate of decrease is exponential in depth under assumed linear independence of inputs. It is followed by Theorem 5, which proves that, under mild smoothness and loss assumptions, the $\textit{expected}$ log norm of gradient of the constructed MLP network, i.e. deep network with linear activations and simplified BN, has finite upper bound $~d^5$.

Following section supports the results by limited experiments, showing that rank collapse does not occur when MLP using (simplified) BN as proposed is used.

Paper follows to discuss possible extensions to treat non-linear networks and proposes activation function shaping through additional pre-activation scale hyperparameter, that is tuned towards zero, effectively linearising explored models using $tanh(\cdot)$ and $sin(\cdot)$ activations.

**Strengths:**

1. To the best of reviewers knowledge presented are novel results (Theorem 1, 5) that are non-asymptotic and hold for networks with finite width, as opposed to previous works. This is very promising and of large interest and value to ML community increasing potential impact of the manuscript.

2. Very well and accessibly written exposition of rather technical methods (random matrix theory, Weingarten calculus) used in proofs.

3. After theoretical results paper (creatively) suggests activation shaping technique in combination with BN increasing and demonstrating practical utility of results.

**Weaknesses:**

#1: While the non-asymptotic and finite width results may be largely impactful as noted in Strengths, the importance may be diminished by applicability on
1. $\textit{expected}$ grad norm
2. grand norm bounds of Theorem 5 may be still prohibitively large $\textit{e^{d^5}}$ and
3. "practicality" may be hindered by "$\textit{linear}$ activation" required (this is somewhat alleviated by last section by introducing more hyper-parameters to bring $sin$ and $tanh$ activations to linear regime).

I believe authors should extend the Discussion or include section to elaborate more on these points. Possibly extend a rather limited experiments to corroborate their arguments in case of a lack of theory.

#2: Technical: Dimensionality discrepancies throughout Section 3, page 3 and 4. For instance:
 - in the definition of $\varPhi(X)$ on page 4, the $\varPhi(X)$ is not function of $R^{d \times d}$.
 - $X$ should be $d \times n$ matrix instead of previously defined $n \times d$ or $XX^T$ should be used instead of $X^TX$

**Questions:**

Q1: Related to Weaknesses, ad 3.) "practicality", Could authors present their view on what benefits are provided by deep linear networks compared to shallower linear nets?

Q2: To address aforementioned limitations of theoretical results (see Weaknesses #1, 1-3),, e.g., to support non-linear activation, authors propose activation shaping combined with use of BN. From their argument it seems that such technique, tuning pre-activation gain $\alpha$ towards zero, effectively linearise the network. Section 5 claims that this technique still maintains the benefits of non-linear activations demonstrated in Fig.5. Could authors elaborate on how exactly are benefits maintained?

Q3: In later sections paper presents intriguing idea of “implicit bias towards orthogonal matrices”, for instance in Fig 6. But one can read experimental results in Fig 6. other way, namely that it suggests that while middle layers have been initialised already “almost orthogonal” their orthogonality gap increased the most during training compared to other layers! Could authors show further evolution of training continued beyond the early stopping or otherwise corroborate "implicit bias" claims more extensively?

---

> ### Author Response · Authors · 2023-11-17
>
> We greatly appreciate the reviewer's thorough evaluation of our work and recognizing the main strengths our non-asymptotic results and practical utility of our activation shaping method.
>
> > While the non-asymptotic and finite width results may be largely impactful as noted in Strengths, the importance may be diminished by applicability on
> > -  expected grad norm
>
> We thank the reviewer for raising this important point.  Fortunately, based on our our non-asymptotic results with exponential behavior, we can use Markov inequality to convert our results in expectation to with high probability results. Assuming same conditions as Thm. 5 and non-degenerate inputs $\phi(X^0)= O(1),$ we can prove that log-gradients are bounded by $O(d^{6})$ with high probabiltiy. Formally, we prove that there is $C$ such that for any $\delta\in(0,1)$
> $$
> P\Big(\log\|\nabla_{W_\ell}\mathcal{L}\|_{op}\ge  C \ln(1/\delta)  d^{6}(\phi(X^0)^3+1)\Big) \le 2\delta
> $$
> Note that this bound has an additional factor of $d^{1},$ due to the fact that we have to ensure that failure events are uniformly bounded across a possibly infinite number of layers.
>
> Please see Appendix H in the updated manuscript for the proof and a detailed discussion of the high probability bound.
>
>
> > - grand norm bounds of Theorem 5 may be still prohibitively large $e^{d^5}$ and
>
> Our main focus was on depth and not network width, denoted by $d$. The main message our paper conveys is the fact that the upper bound in Thm. 5 holds for an *arbitrarily deep* network. Thus, the main take away message from Thm. 5 is that a BN net with random orthognal weights has bounded gradients. Remarkably, [1] establish that gradient norms grow at exponential rate with depth. Our results shows that this exponential depedency is replaced with constant if we use orthogonal random weights. Yet, we agree with the reviewer that finding the tighter bound for width could be interesting for follow-up studies.
>
>
> > - "practicality" may be hindered by "activation" required (this is somewhat alleviated by last section by introducing more hyper-parameters to bring *sin* and *tanh* activations to linear regime).
>
> One of the main practical outcomes of our work is the activation shaping mechanism, showing that gradient explosion in a network with BN can be avoided using orthogonal weights rather than Gaussian. Note that both practically and theoretically, gradient explosion is "unavoidable" using Gaussian weights [1]. To our knowledge, the proposed approach is the only existing practical solution to avoid gradient explosion in a deep network with BN.
>
>
> > - I believe authors should extend the Discussion or include section to elaborate more on these points. Possibly extend a rather limited experiments to corroborate their arguments in case of a lack of theory.
> >
> We have added Section H in the Appendix. Furthermore, experiments presented in Sections 4, 5 and 6 support our theoretical results: going beyond the initialization (Section 4, 6), using non-linear activations (Sections 4, 5, 6), and using standard BN (Sections 4, 5, 6). Furthermore, we explore various depths and datasets for training, showing that the empirical observations are not ad-hoc to a particular data distribution or network configuration. Please let us know whether the above response addresses your comments. We can gladly provide further details.
>
>
> > 2.  Technical: Dimensionality discrepancies throughout Section 3, page 3 and 4. For instance: ...
>
> While the $n=d$ assumption in our analysis makes these notations equivalent, we have nevertheless corrected the notations to avoid any confusions (Section 3.1, paragraph "Isometry gap").

---

> ### Author Response · Authors · 2023-11-17
>
> > - Q1: Related to Weaknesses, ad 3.) "practicality", Could authors present their view on what benefits are provided by deep linear networks compared to shallower linear nets?
>
> Our analysis in Theorem 1 proves that deep networks with batch normalization and linear activations are not linear. In particular, these networks can makes their output orthogonal as they grow with depth. This is due to the non-linearity introduced by the batch normalization operator.
>
> Networks with linear activations and BN are a simplified model used to study the problem of gradient explosion, as these networks also suffer from gradient explosion (Yang & Pennigton et al. 2019). First, we theoretically study these simple models. Based on insights from our theoretical analysis, we propose practical recipes: orthogonal initialization and activation function shaping. Specifically, we use three main insights from networks with linear activations: initialization scheme, shaping mechanism and the choice of batch size for BN.
>
> > - Q2: ... Section 5 claims that this technique still maintains the benefits of non-linear activations demonstrated in Fig.5. Could authors elaborate on how exactly are benefits maintained?
>
> Plese note that Figures G1-G2 show that the shaped $\sin$ and $\tanh$ activations have better accuracy than the linear case, and hence substantiate that the non-linearity in the activations is "effectively" used.
>
>
> > - Q3: In later sections paper presents intriguing idea of “implicit bias towards orthogonal matrices”, for instance in Fig 6. But one can read experimental results in Fig 6. other way, ...
>
>
> Let us stress that all fully connected weights are initialized to *perfect orthogonality* i.e. $WW^T = W^TW = I$, implying that at initialization the weights have zero isometry gap $\phi(W_\ell)=0$ for all $\ell.$ This would correspond to a flat $\phi(X_\ell)=0$ curve for the $0$th epoch, which is not shown in Figure 6. Furthermore, Figure 6 shows that the middle layers throughout training have lower isometry gap, as evidenced by the U-shaped curves in all epochs. This U-shape indicates that the amount of change in orthogonality, as measured by the isometry gap, is smaller in the middle layers, in contrast with the first and last layers. Thus, if we track the change in orthogonality between epoch $0$ up to epochs $50,100,200,300,$ the middle layers have much smaller change in isometry gap compared to first and last layers.
>
> >  Could authors show further evolution of training continued beyond the early stopping or otherwise corroborate "implicit bias" claims more extensively?
>
> Our results are not achieved for early stopping as we have trained the networks for 300 epochs. To supplement our observations, we provided Figures G4 where we observe 300 epochs suffice to acheive high training and test accuracies.
>
> [1] Yang, Greg, et al. "A mean field theory of batch normalization." arXiv preprint arXiv:1902.08129 (2019).

---

> > ### Comment · Reviewer_GLi5 · 2023-11-23
> >
> > I thank authors for addressing raised concerns and doing suggested edits. In my opinion overall, the paper provides a valuable contributions with few minor gaps remaining to be closed, e.g., Q3 response and corroborating a bias towards orthogonal matrices (training for a given fixed number, be it 300, epochs, is a heuristic regularisation falling under "early stopping" label. What is "early" anyways? :-)). I keep my rating (weak accept). Thanks again for your contributions.

---

### Official Review · Reviewer_eDWh · 2023-10-31

**Soundness:** 3 good
**Presentation:** 3 good
**Contribution:** 3 good
**Rating:** 6
**Confidence:** 3

**Summary:**

The authors address the problem of gradient explosion with depth in networks with Batch Normalization (BN) outlined by Yang et al. [1]. They show that infinitely deep linear networks can be trained with BN while avoiding the problem of gradient explosion as long the weights are orthogonal.

[1] Yang, Greg, et al. A Mean Field Theory of Batch Normalization. International Conference on Learning Representations. 2018.

**Strengths:**

1. The authors present a proof for avoiding gradient explosion in infinitely deep linear networks with BN as long as the inputs are full rank and the weights of the network are orthogonal.

2. In linear MLPs with upto 1000 layers, they show that orthogonality is approximately maintained in the middle layers as required by the proposed theorem.

3. The presented proof and argument is easy to follow.

**Weaknesses:**

1. The authors have not included literature on recent work on inducing dynamical isometry in feedforward networks which has shown to also avoid the need for BN. The discussion seems relevant for this paper (see [1], [2], [3]).

2. The proof relies on the fact that the network is has linear activations and the weights are orthogonal during training. Although, this seems to somewhat hold empirically for linear networks, how reasonable is it to expect such a condition to hold for nonlinear activations. The weights will likely not stay orthogonal during training for nonlinear activations. Further, can the analysis be extended to nonlinear ReLU - like activations, by potentially using a first order approximation of the ReLU [4] or other approximations?

3. Since the proof requires for the network weights to be orthogonal, the network is only restricted to learn a rotation of the inputs. This seems restrictive from an expressiveness point of view unless I misunderstand something.

[1] Rebekka Burkholz and Alina Dubatovka. Initialization of relus for dynamical isometry. Advances in Neural Information Processing Systems, 2019.

[2] Yaniv Blumenfeld, Dar Gilboa, and Daniel Soudry. Beyond signal propagation: is feature diver-
sity necessary in deep neural network initialization? In International Conference on Machine
Learning, 2020

[3] Andrew Brock, Soham De, and Samuel L. Smith. Characterizing signal
propagation to close the performance gap in unnormalized ResNets. International Conference on Learning Representations. 2020.

[4] Rebekka Burkholz. Most activation functions can win the lottery without excessive depth. Advances in Neural Information Processing Systems 2022.

**Questions:**

See above section for questions.

---

> ### Author Response · Authors · 2023-11-17
>
> We highly appreciate the reviewer for pointing us to further related works on dynamical isometry. We have included and discussed these valuable references in the Related Work part of our paper, paragraph "The challenge of depth in learning".
>
> > 3. Since the proof requires for the network weights to be orthogonal, the network is only restricted to learn a rotation of the inputs. This seems restrictive from an expressiveness point ...
>
>  Theorem 1 proves that the network does more than a rotation: the network makes outputs orthogonal for each non-degenerate input matrix. While the rotation operation is angle preserving, orthogonalization increases the angle between every pair of vectors. Indeed, batch normalization, as a non-linear function, makes the input-output function siginficantly non-linear as the network grows in depth. Empirical observations show that networks with batch normalization and random weights become more expressive by increasing depth [1].
>
> > 2. The proof relies on the fact that the network is has linear activations and the weights are orthogonal during training. ...
>
> Having orthogonal weights is not only an assumption for theoretical analysis, but it is necessary to avoid gradient explosion at initialization. For example, [2] proves that the intialization with Gaussian weights causes gradient explosion in depth. Avoiding gradient explosion for batch normalization has been an open problem since 2015 and the literature concluded that the gradient explosion is "unavoidable" with batch normalization [2]. Our study starts with a theoretical setting where the activation function is linear. Taking inspiration from our analysis, we propose an activation shaping mechanism and initializing the weights with orthogonal matrices in order to avoid the gradient explosion for networks with non-linear activations.
>
> Please note that the established analysis is at *initialization,* when the weights are random. There is no assumption on the training dynamics in the established analysis. However, we observe the weights remain almost orthogonal during training in practice (see Section 6).
>
>
> > Further, can the analysis be extended to nonlinear ReLU - like activations, by potentially using a first order approximation of the ReLU [4] or other approximations?
>
>
> We only considered sine and hyperbolic tangent activations, as proof-of-concept for an activation shaping scheme inspired by our theoretical analysis. For ReLU networks, one can modify the shaping mechanism by introducing two parameters, the gain $\alpha$ and the bias $\beta$ as $\sigma(\alpha (x - \beta) )$, towards avoiding gradient explosion in depth.
>
> [1] Frankle, Jonathan, David J. Schwab, and Ari S. Morcos. "Training batchnorm and only batchnorm: On the expressive power of random features in cnns." arXiv preprint arXiv:2003.00152 (2020).
>
> [2] Yang, Greg, et al. "A mean field theory of batch normalization." arXiv preprint arXiv:1902.08129 (2019).

---

> > ### Comment · Reviewer_eDWh · 2023-11-20
> > **Response to Rebuttal**
> >
> > I thank the authors for their clarifications and I am satisfied with their response.
> >
> > Just a follow up question, while the Batch Norm parameters ($\gamma, \beta$) are shown to be expressive if the network is deep enough, the networks required for training only BN would be considerably deeper (and wider) than traditional networks.
> > Does this suggest a tradeoff between expressiveness of the network and eliminating gradient explosion by way of orthogonal weights in the proposed theoretical setting?

---

> > > ### Author Response · Authors · 2023-11-21
> > > **Response to follow up question**
> > >
> > > We thank the reviewer very much for this interesting question. The tradeoff between expressiveness and trainability in very deep networks is an interesting topic to study. To the authors' knowledge, there is currently no theoretical quantitative result to characterize this tradeoff. However, this is a very promising avenue for future research that we will add in the discussions.

---

### Official Review · Reviewer_692A · 2023-11-01

**Soundness:** 3 good
**Presentation:** 3 good
**Contribution:** 3 good
**Rating:** 6
**Confidence:** 3

**Summary:**

The paper shows the existence of a batch-normalized network that avoids rank collapse and does not suffer from exploding gradients as depth increases. To construct a network that satisfies the above properties, the paper uses an initialization scheme where the weights are random orthogonal matrices.

**Strengths:**

The isometry and log gradient norm bounds are non-asymptotic bounds. Furthermore, the usage of Weingarten calculus to obtain rates for isometry increase is interesting.

**Weaknesses:**

The paper uses a modification of batch normalization (in particular the mean centering operation is removed) when proving bounds on the isometry gap and log gradient norms (however, the experiments seem to suggest that this is not an issue).

The paper also considers the setting where the number of training examples is equal to the dimension of the problem..

**Questions:**

Do you obtain figures similar to F1 when using ReLU activations?

---

> ### Author Response · Authors · 2023-11-17
>
> We greatly appreciate the reviewer's insightful comments, as well as the recognition of the main strengths of our work, namely the non-asymptotic bounds and the usage of Weingarten calculus.
>
> >  The paper also considers the setting where the number of training examples is equal to the dimension of the problem.
>
> We thank the reviewer for pointing out this potentially confusing issue. We assume the *mini-batch* size is equal to the width. However, the number of samples in the training dataset can be arbitrarily large. Notably, batch normalization layers use the statistics of intermediate representations for input batches. The batch size is a hyperparameter that can be adjusted to achieve a better performance. Here, we suggest a precise value for this hyperparameter to avoid the issue of gradient explosion.
>
> We have made the distinction between number of training samples and mini-batch size clear in the revised text, in Section 3, Equation 3.
>
> >  The paper uses a modification of batch normalization (in particular the mean centering operation is removed) when proving bounds on the isometry gap and log gradient norms (however, the experiments seem to suggest that this is not an issue).
>
> We confirm the BN modification (no centering) was only used for the theoretical setting, and all the empirical results shown in Sections 4, 5, and 6 use standard BN. Remarkably, this simplification still provides insights on the standard BN. Simplified versions of BN have been used in prior theoretical analyses and empirical research [1, 2].
>
>
> > - Do you obtain figures similar to F1 when using ReLU activations?
>
> The reviewer is raising an interesting and important question. The proposed activation shaping scheme, which allows us to avoid gradient explosion for sine and hyperbolic tangent, relies on the fact that they are both differentiable at $x=0.$ Intuitively, when $\sigma'(0)$ exists, we can linearize $\sigma(\alpha x )$ more accurately when $\alpha$ is smaller. This property breaks down for ReLU, which is not differentiable at the origin. It is certainly worth exploring in a follow-up study how to extend the proposed activation shaping scheme for non-differentiable functions like ReLU.
>
> [1] Daneshmand, Hadi, Amir Joudaki, and Francis Bach. "Batch normalization orthogonalizes representations in deep random networks." Advances in Neural Information Processing Systems 34 (2021): 4896-4906.
>
> [2] Salimans, Tim, and Durk P. Kingma. "Weight normalization: A simple reparameterization to accelerate training of deep neural networks." Advances in neural information processing systems 29 (2016).

---

### Author Response · Authors · 2023-11-17

We sincerely appreciate all the reviewers' time and effort to evaluate our work, and help us improve our paper. We thank reviewers `GLi5` and `uof1` for recognizing the potential impact of our work:

> This is very promising and of large interest and value to ML community increasing potential impact of the manuscript.

> Normalization is used everywhere in ML, so studying it has a high impact.

This excerpt from reviewer `GLi5` has summarized our contribution excellently:

> The paper addresses limits of gradient descent loss minization of (very) deep networks using batch normalization (BN).

Furthermore, the non-asymptotic nature of our results are considered a strong theoretical contribution by reviewers `692A`, `GLi5`, `uof1`, as captured in the following excerpt:
> The isometry and log gradient norm bounds are non-asymptotic bounds. Furthermore, the usage of Weingarten calculus to obtain rates for isometry increase is interesting.

From a different perspective, Both reviewers `692A` and `GLi5` acknowledge the creative use of Weingarten calculus in our work, while keeping it accessible to read:
> Very well and accessibly written exposition of rather technical methods (random matrix theory, Weingarten calculus) used in proofs.


Besides our theoretical contributions, reviewers `uof1` and `GLi5` credit the practical benefits of our activation shaping scheme, as shown in this excerpt:
> After theoretical results paper (creatively) suggests activation shaping technique in combination with BN increasing and demonstrating practical utility of results.

We also acknowledge the issues raised by the reviewers and have taken steps to address them by providing point-by-point responses. Since some points were raised by multiple reviewrs, we include a common response to them here in the general response:

- BN without mean reduction

While this simplification is done solely in the theoretical part of the manuscript, we show experimentally that our result still holds even when using the standard BN.

- Concerns on the practicality of linear activations

Similarly, the linear activation functions are used in order to rigorously prove the main theorems of the manuscript. Notably, networks with linear activations and BN also suffer from gradient explosion, as proven by [1]. Based on the theoretical insights obtained from linear activations, we derive an activation shaping mechanism that empirically expands our result to non-linear activations.

We have uploaded a new manuscript with all the changes highlighted in blue to address the reviewers' comments.

[1] Yang, Greg, et al. "A mean field theory of batch normalization." arXiv preprint arXiv:1902.08129 (2019).

---

### Meta-Review · Area_Chair_Vp2Q · 2023-12-23

**Metareview:**

The paper is well-received by the reviewers for its study of batch normalization in deep network training. Reviewers emphasize its non-asymptotic analysis compared to prior works, and the use of Weingarten calculus. Practical aspects, like the activation shaping technique alongside BN, are also appreciated.  There are a few limitations of the work, such as only studying linear activations and the high dependence on the width. Nevertheless, I think that these limitations are outweighed by the contributions. Overall, while it is not clear what practical impact this paper can have, I think the paper does improve our understanding of batch normalization in deep neural networks. I recommend acceptance.

**Justification For Why Not Higher Score:**

The study is limited to linear activations, and orthogonal initialization. This makes the analysis limited.

**Justification For Why Not Lower Score:**

The paper does analyze a new setting (non-asymptotic analysis of batch normalization with multiple layer), in which batch normalization does not have gradient explosion issue. This is a useful addition to the literature.

---

### Decision · Program_Chairs · 2024-01-16

Accept (poster)